# CoRe-GD: A Hierarchical Framework for Scalable Graph Visualization with GNNs

**Florian Grötschla, Joël Mathys, Robert Veres & Roger Wattenhofer**
ETH Zurich
`{fgroetschla,jmathys,rveres,wattenhofer}@ethz.ch`

## Abstract

Graph Visualization, also known as Graph Drawing, aims to find geometric embeddings of graphs that optimize certain criteria. Stress is a widely used metric; stress is minimized when every pair of nodes is positioned at their shortest path distance. However, stress optimization presents computational challenges due to its inherent complexity and is usually solved using heuristics in practice. We introduce a scalable Graph Neural Network (GNN) based Graph Drawing framework with sub-quadratic runtime that can learn to optimize stress. Inspired by classical stress optimization techniques and force-directed layout algorithms, we create a coarsening hierarchy for the input graph. Beginning at the coarsest level, we iteratively refine and un-coarsen the layout, until we generate an embedding for the original graph. To enhance information propagation within the network, we propose a novel positional rewiring technique based on intermediate node positions. Our empirical evaluation demonstrates that the framework achieves state-of-the-art performance while remaining scalable.

## 1 Introduction

Graphs, fundamental structures in discrete mathematics, are ubiquitous in various fields. Visualizing graphs as node-link diagrams, where nodes are represented as points or circles and edges as connecting lines, is a common practice. The quality of visualizations plays a crucial role in comprehending the underlying graph structures. For example, when represented in a good way, the underlying structure of the graph in Figure 1 becomes clear. A prevalent approach to enhancing graph visualizations involves the optimization of a stress function, trying to arrange the nodes such that the Euclidean distance between any two nodes closely approximates their shortest path distance in the graph. However, achieving optimal layouts presents a substantial computational challenge (Demaine et al., 2021). To address this challenge, these algorithms often resort to heuristic methods. For instance, force-directed algorithms conceptualize the graph as a physical system governed by forces akin to springs or magnetic fields (Kamada & Kawai, 1989). These algorithms simulate the system iteratively to optimize graph stress, allowing nodes to reach equilibrium positions.

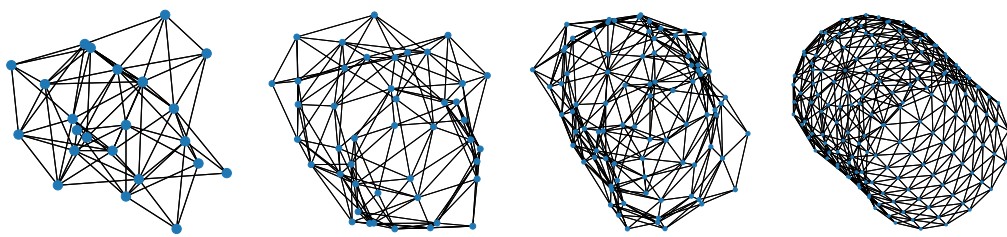

Figure 1: Graph Evolution with CoRe-GD: From coarse to fine. Our hierarchical approach prioritizes global positioning before local optimization. The resulting visualization makes the underlying graph understandable. Graphs were drawn with CoRe-GD model trained for 3 dimensions.

Tailoring heuristics to specific graph distributions is often necessary to produce high-quality visualizations. This requirement frequently demands domain-specific expertise but also provides an avenue for learning-based solutions. Recent research (Giovannangeli et al., 2021; Wang et al., 2021) has drawn parallels between force-directed algorithms and message-passing graph neural networks, where nodes communicate with their neighbors. Nonetheless, scalability remains a challenge. When presented with larger graphs necessitating more global context, these methods do not scale. While extending message-passing to fully connected graphs can address this limitation, it incurs quadratic computational and message complexity in the number of nodes, rendering it computationally impractical for large graphs. We therefore propose a new framework with a focus on scalability: **CoRe-GD**, a scalable **Co**arsening and **Re**wiring Framework for **G**raph **D**rawing. CoRe-GD combines hierarchical graph coarsening with a novel positional rewiring technique, facilitating communication beyond local neighborhoods. Notably, both coarsening and rewiring are computationally efficient and introduce only a linear message-passing overhead.

Our approach begins with a hierarchical coarsening process, generating multiple levels. At the coarsest level, nodes from the original graph are merged into supernodes, capturing global structural properties. We prioritize optimizing the positions of these supernodes as an initial step. Subsequently, we reverse the coarsening process by uncontracting the supernodes, allowing us to refine the local placements. This iterative refinement continues until we reach the finest coarsening level, which corresponds to the original graph. This global-first strategy enables us to generate rough initial graph drawings and then iteratively optimize them, all while keeping essential information pathways short and effective. Within each coarsening level, we employ Graph Neural Networks (GNNs). To enhance communication between nodes that might be far apart in the original graph but are close in the current visualization, we introduce additional message-passing steps. This adjustment is made because nodes placed far apart in the visualization, despite being nearby in the original graph, can lead to a loss in quality due to the discrepancies between their shortest-path distances and their current spatial separation. Therefore, these supplementary message-passing steps facilitate better message flow to improve the layout.

Our contributions can be summarized as follows:

1. We introduce CoRe-GD, a scalable neural framework for graph visualization based on coarsened graph hierarchies.

2. We present a novel positional rewiring technique that leverages intermediate decoded embeddings for better information flow.

3. We perform extensive experiments on various datasets, showcasing state-of-the-art performance, even compared to sophisticated handcrafted algorithms.

## 2 RELATED WORK

**Graph Drawing.** Given an undirected and connected graph $\mathcal{G} = (V, E)$, we are concerned with finding an embedding $\Gamma : V \rightarrow \mathbb{R}^d$ such that $\Gamma(v)$ represents the position of node $v$ in a node-link diagram. In the graph drawing literature, many metrics have been introduced to judge the performance of these layouts (Purchase, 2002). One commonly used formulation is the notion of stress, which can be traced back to Kruskal (1964) and was first used by Kamada & Kawai (1989) to generate aesthetically pleasing layouts for the class of general graphs. It is defined as:

$$\text{stress}(G, \Gamma) := \sum_{u,v \in V, u \neq v} w_{uv}(\|\Gamma(u) - \Gamma(v)\|_2 - d_{uv})^2$$

where $d_{uv}$ refers to the shortest path distance between node $u$ and $v$, $\|x\|_2$ is the Euclidean norm of $x$, and $w_{uv} := d_{uv}^{-2}$. Here, the graph can also be interpreted as a physical system with springs between all node pairs that pull or push nodes apart such that an equilibrium is reached when their distance is close to their shortest path distance in the graph. The algorithm falls into the category of *force-directed* graph drawing, which was previously popularized by Eades (1984) and further studied by Fruchterman & Reingold (1991). These algorithms can be improved using a variety of heuristics (Frick et al., 1995). Another line of research uses multi-dimensional scaling (Torgerson, 1952) for graph drawing, improved upon by using landmarks (Silva & Tenenbaum, 2002) and pivots (Brandes & Pich, 2007). De Leeuw (1988) introduced majorization to tackle the scaling problem and Gansner et al. (2005) used it directly for graph drawing. More recently, Ahmed

et al. (2020) used Gradient Descent on the stress function, with Ahmed et al. (2022) extending the approach to stochastic GD. For very large graphs, maxent-stress was proposed by Gansner et al. (2012). Meyerhenke et al. (2017) use this to draw very large graphs based on a multi-level coarsening, which largely motivates our approach. The results for classical algorithms motivated a line of learning-based techniques. Wang et al. (2020) use a bidirectional graph LSTM to process the graph, while Giovannangeli et al. (2021) use a U-net like architecture. Other approaches use a fully connected graph where edges are annotated with precomputed shortest path distances between node pairs (Wang et al., 2021), positional encodings together with a Graph Neural Network (Tiezzi et al., 2022) or Generative Adversarial Networks to learn from examples (Wang et al., 2022b).

**Graph Neural Networks.** Introduced by Scarselli et al. (2008), Graph Neural Networks (GNNs) have become state-of-the-art for many tasks in graph learning. There are two common problems with GNNs that follow the original purely message-passing-based approach: Their expressivity is bounded by the Weisfeiler-Lehman (WL) algorithm (Xu et al., 2018), and message-exchange over many rounds is susceptible to oversmoothing (Chen et al., 2020) and oversquashing (Alon & Yahav, 2020). To address the first problem and to make GNNs more expressive, one can endow nodes with additional features such as random bits (Sato et al., 2021) or subgraph isomorphisms (Bouritsas et al., 2022). Another solution to facilitate information exchange between all nodes, even without using many rounds, is to allow message exchange between all nodes in the last layer (Alon & Yahav, 2020). For some architectures, positional encodings are used to give nodes a relative sense of where they are located. They range from Laplacian PE's (Wang et al., 2022a; Dwivedi et al., 2022), shortest path encodings (Li et al., 2020; Ying et al., 2021) and random walks (Dwivedi et al., 2021) to structural features (Chen et al., 2022). To further improve the flow of messages in the graph and alleviate the aforementioned problems, rewiring approaches change the topology of the graph that is used for message-passing. Metrics such as the Ricci flow (Topping et al., 2021) or spectral features (Koutis & Le, 2019) can be used to decide which edges to rewire. Pei et al. (2020) propose a learnable module that determines the rewired edges based on distances in a latent embedding space. Furthermore, graph coarsening approaches have also been used for GNNs (Huang et al., 2021).

## 3 THE CORE-GD FRAMEWORK

Figure 2 provides an overview of the CoRe-GD architecture. CoRe-GD employs a coarsening hierarchy to optimize layouts, starting from coarser representations and progressing to the original graph. At each level, a recurrent layout optimization module refines the layout via positional rewiring.

**Hierarchical Optimization.** The coarsening hierarchy (see Section 3.2) is computed to generate a series of graphs with fewer nodes and projections from coarser to finer levels. On the coarsest graph, node initialization is performed (see Section 3.1), followed by embedding through an encoder network. These embeddings undergo layout optimization that improves the positioning of nodes, and the refined embeddings are transferred to the next level. The process continues until the finest level is reached, representing the original graph, where a final layout optimization is conducted, and node embeddings are decoded to positions.

**Layout Optimization with Positional Rewiring.** Nodes in the graph that are connected over many hops can lead to high stress when they are positioned too close to each other. As these nodes need many message-passing rounds to communicate with each other, we rewire the graph and add edges between nodes in close proximity. We focus on efficient rewiring methods, adding only a linear overhead in message-passing complexity. We experiment with three rewiring techniques: (1) K-Nearest-Neighbor graphs, (2) Delaunay triangulations (Delaunay et al., 1934), and (3) Radius graphs. These techniques are computationally efficient, with (1) and (2) introducing only a linear number of edges. In Figure 2 (bottom part), we exemplify the utility of the rewiring for the node surrounded by a dotted cycle. This node is close to a node below it but not directly connected to it. For a stress-optimized drawing, where distances should approximate shortest path distances, this leads to a high loss. However, in the rewiring that was computed based on the closeness of nodes, they become connected, letting them exchange messages through a differently parametrized graph convolution $Conv_R$. More motivation and examples for the rewiring can be found in Appendix A.3. The output is then passed to the convolution on the original topology $Conv_E$ again, and this alternation of layers continues for $r$ rounds. Notably, the same decoder is used to predict final positions,

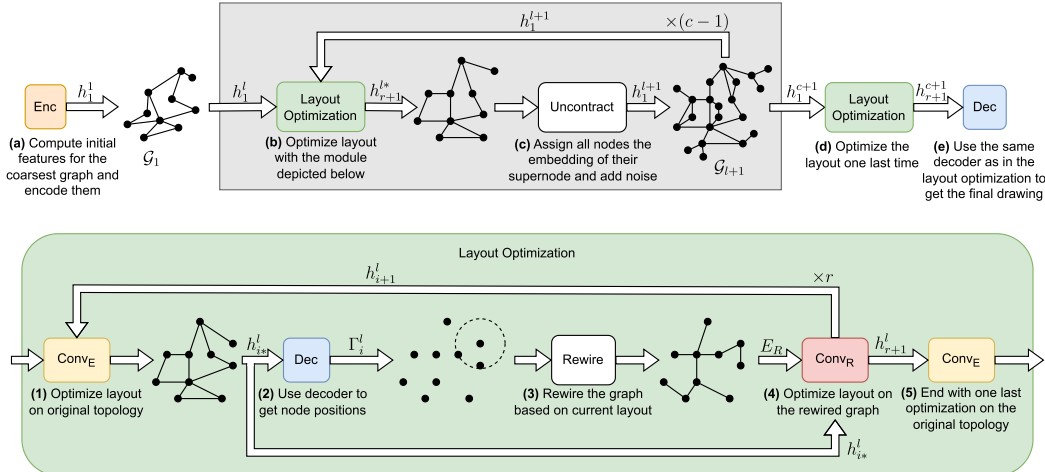

Figure 2: Architecture overview of CoRe-GD: **On top: (a)** An encoder creates initial embeddings for nodes on the coarsest level (see Section 3.1). **(b)** The embeddings are then successively refined in the *layout optimization* module (depiction below) before the graph is **(c)** uncontracted and embeddings are projected to the new nodes (see Section 3.2). After $c - 1$ repetitions (with $c$ being the number of coarsening levels), the original graph is recovered, and **(d)** the layout is optimized one final time before **(e)** being decoded into the final node positions. **On the bottom:** Overview of the layout optimization: To refine the latent embeddings of a given graph, we **(1)** execute a GNN convolution on the original topology. **(2)** The resulting embeddings are then decoded into node positions that undergo the rewiring procedure **(3)**, resulting in a new set of edges $E'$ for the rewired graph. **(4)** These edges are then used for another GNN convolution on $E'$ to enhance information exchange between far-away nodes. The two convolutions are alternated, and the rewiring is re-computed $r$ times before **(5)** one last convolution on the original topology is applied.

and while $\text{Conv}_\text{E}$ and $\text{Conv}_\text{R}$ do not share their parameters, there only exists one parametrization of each that is applied recurrently. The GNN convolutions make use of a Gated Recurrent Unit (GRU) (Huang & Carley, 2019; Grötschla et al., 2022) that was shown to be beneficial for recurrent graph neural networks. Moreover, the convolutions incorporate the embeddings of both endpoints $u$ and $v$ when sending a message along the edge $(u, v)$. For a formal definition of the tested graph convolutions, we refer to Appendix A.2.1. The embeddings $h$ that are indexed in both figures are consistent in their usage with the provided pseudocode for both modules (see Appendix A.2).

## 3.1 Node Initialization

To motivate the use of initial features, we first analyze the requirements for Graph Drawing in terms of WL-expressiveness. The following lemma provides a first insight:

**Lemma 1** *To minimize stress for arbitrary graphs, the distinguishability power for nodes has to exceed that of the $k$-WL algorithm for any $k$.*

This follows directly from the fact that there exist graphs with several nodes in the same orbit, i.e., in the same equivalence class under graph automorphism. These nodes can not be distinguished by $k$-WL for any $k$ as they are structurally the same. One example is cycle graphs, where all nodes are part of the same orbit. Therefore, a GNN that is limited by any $k$-WL algorithm will map these nodes to the same embeddings and thus node positions (if we assume that node positions are computed from the node embeddings directly). For a cycle graph, this is clearly not desirable and will end in a drawing that is far from optimal. A common way to enrich existing architectures with more expressive power is adding node features, for example, in the form of positional or structural encodings (Rampášek et al., 2022). These features not only help to distinguish nodes but also provide helpful information on the graph topology that can help solve the task at hand. Thus, CoRe-GD initializes node embeddings with a combination of features. While the framework allows for the use of

any features, we further choose initializations that have been proven to work well in existing Graph Drawing algorithms and models. We use (1) Laplacian Positional Encodings, (2) Distance encodings to beacon nodes, and (3) random features. Random features and laplacian PE's have been used in prior work for graph drawing (Ahmed et al., 2022; Tiezzi et al., 2022). The reasoning for the usage of distance encodings is that while knowing all-pairs-shortest-path distances would be preferred and has been shown to be generally useful to enhance the performance of GNNs (Zhang et al., 2023), the computation and memory overhead make them infeasible for large instances. Knowing the distances to a constant number of nodes in the graph can serve as a surrogate (Ahmed et al., 2022). This idea has been shown to be effective for graph drawing, e.g., in Landmark MDS (Silva & Tenenbaum, 2002) or Pivot MDS (Brandes & Pich, 2007), and similar ideas were used for GNNs (You et al., 2019). More formally, we choose $n_b$ beacon nodes $V_b = \{v_b^0, \ldots, v_b^{n_b-1}\}$ uniformly at random and compute $d_{u,v}$ for every node $u$ to every beacon $v$ using separate breadth first searches. Once computed, we assign every node $u$ a vector with distances to all beacon nodes $(d_{u,v_b^0}, \ldots, d_{u,v_b^{n_b-1}})^T$.

These distance values are represented using sine-cosine positional encodings in a transformer-style fashion (Vaswani et al., 2017) and can thus be seen as a generalization of sequential positional encodings. In general graphs, a single beacon may not suffice to uniquely identify each node. Previous work on routing protocols (Wattenhofer et al., 2005) provides an analysis for selecting the necessary number and type of nodes in various graph classes to ensure unique identification. We employ random selection to minimize additional overhead and recompute all features between training epochs.

## 3.2 GRAPH COARSENING

The coarsening hierarchy aims for a global-first layout optimization strategy before transitioning to progressively finer local placements. Beginning with the original graph $\mathcal{G} = (V, E)$, the coarsening process generates a partition $P = p_1, \ldots, p_k$ of the node set $V$. This partitioning optimizes specific criteria, such as edge-cut minimization or graph spectrum preservation (Jin et al., 2020). It results in a new graph $\mathcal{G}' = (V', E')$, where nodes $v' \in V'$ correspond to partition elements $p \in P$. A mapping function $f : V \to V'$ assigns nodes to their respective partition (termed "supernode"). This coarsening can be iteratively applied until the graph becomes sufficiently small. In our experiments, we employ spectral-preserving coarsening (Jin et al., 2020) with a constant reduction factor to yield a sequence of coarsened graphs $\mathcal{G}_1, \ldots, \mathcal{G}_c$, where $\mathcal{G}_c$ matches the original graph. This hierarchy requires approximately $c \in \mathcal{O}(\log |V|)$ levels. In the CoRe-GD framework, layout optimization begins at the coarsest level $\mathcal{G}_1$ before progressing to finer levels. Transitioning from layer $i$ to $i + 1$ we apply the inverse mapping $f^{-1}$ to the embeddings of the current graph $\mathcal{G}_i$, i.e., for nodes within the same supernode, we set the finer layer's embeddings to that of the coarser layer's supernode: $h_{(v)}^{l+1} = h_{(f^{-1}(v))}^l$, for every node $v \in V$. To distinguish these nodes further, we randomly sample noise from a normal distribution and add it to the embeddings.

## 3.3 TRAINING

**Scale-Invariant Stress.** Previous works (Wang et al., 2021; Giovannangeli et al., 2021) apply the stress metric directly as the loss for self-supervised training, which we find to have difficulties with when working with graphs of varying sizes. Especially when scaling to bigger instances, graphs might need a larger drawing area than smaller ones. However, for training our model, we prefer for the generated values to stay within the same numerical range. Therefore, we restrict the layout to $[0, 1]^d$ by applying a sigmoid activation. Unfortunately, this has the adverse effect that stress optimal layouts can have different geometries when restricted to this area: Consider the path graph with three nodes as depicted in Figure 3. When restricted to $[0, 1]^2$, the optimal layout has a 90-degree angle at the center node, while the optimal layout that can be achieved in the unrestricted domain aligns all nodes on a line. Thus, we *rescale* node positions by finding the optimal scaling factor $\alpha$ for node positions $P$, such that the scaled layout $P_\alpha := \{\alpha \cdot p \mid p \in P\}$ has optimal stress among all $\alpha$. This allows us to predict values in $[0, 1]^d$ that represent an optimal layout when scaled to a larger area. In this sense, the drawings we generate are *scale-invariant*. The following lemma provides a closed-form solution. A proof can be found in Appendix A.5.

**Lemma 2** *The scaling factor $\alpha$ can be calculated in a closed-form solution:*

$$\alpha_{G,\Gamma} = \frac{\sum_{u \neq v} w_{uv} \|\Gamma(u) - \Gamma(v)\|_2 d_{uv}}{\sum_{u \neq v} w_{uv} \|\Gamma(u) - \Gamma(v)\|_2^2}$$

*and is unique, as the stress is convex with regard to the scaling factor.*

**Replay Buffer.** To incentivize steady improvements for every round of layout optimization, we randomize the number of rounds that we execute before backpropagating the loss through time (Werbos, 1988) during training. We thus avoid training the network for the number of rounds that are used during inference, as training deep GNNs tends to become unstable and requires a higher memory footprint. Instead, we use a *replay buffer*, similar to what is commonly used in Reinforcement Learning (Lin, 1992). The buffer contains a fixed number of graphs and their respective node embeddings that were generated during previous runs. We then interleave batches from the original dataset and batches from the replay buffer during training. After training a complete batch of graphs, every graph and its generated embeddings can randomly replace an element in the buffer. We store latent embeddings in the buffer instead of positions to seamlessly continue the training from a previous state in the execution. By doing so, we only have to backpropagate the loss through a few rounds at a time but expose the network to embeddings that were generated after more rounds during the training. While replay techniques have been used for GNNs before (Grattarola et al., 2021), to the best of our knowledge, we are the first to use intermediate latent embeddings. With this method, our trained models used in Section 4 run hundreds of GNN-convolutions without destabilizing. More details and pseudocode for the training can be found in Appendix A.2.

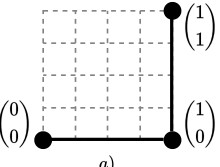

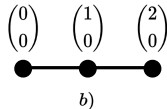

Figure 3: Part a) shows the best drawing inside a $[0,1]^2$ bounding box and b) one best drawing without restrictions.

### 3.4 TIME COMPLEXITY

CoRe-GD was designed with scalability in mind. Apart from an efficient and practical implementation, good time-complexity bounds are thus crucial, especially for larger instances. We show that the complexity of our instantiation is sub-quadratic.

**Lemma 3** *Let $T_{coarsen}, T_{initial}$ and $T_{rewire}$ be the runtime complexities for coarsening, initial feature computation and rewiring of a connected graph $\mathcal{G} = (V, E)$. Further, let $E_{rewire}$ be the upper bound for the number of edges added through the rewiring. Then the following time complexity upper-bound holds for CoRe-GD without hierarchical coarsening:*

$$T_{CoRe\text{-}GD} \in \mathcal{O}\Big(T_{initial}(|V|,|E|) + T_{rewire}(|V|) + E_{rewire}(|V|) + |E|\Big)$$

*For CoRe-GD with hierarchical coarsening, it becomes:*

$$T_{CoRe\text{-}GD}^{hierarchical} \in \mathcal{O}\Big(T_{coarsen}(|V|,|E|) + \log|V| \cdot \big(T_{rewire}(|V|) + E_{rewire}(|V|) + |E|\big)\Big)$$

Notice that the term for computing the initial features disappears if we use a hierarchical coarsening as the initial features are only computed on the coarsest graph, which has a constant size. This has the practical implication that features requiring more heavy precomputation can be used with the hierarchical approach. The log term stems from the number of coarsening levels.

We now turn our attention toward the instantiations of the CoRe-GD framework we use in our experiments. Namely, we use laplacian PEs, beacon distances, random features as initializations, the coarsening borrowed from Jin et al. (2020) to compute the hierarchy, and k-nearest-neighbor rewiring. Laplacian PEs can be computed in $\mathcal{O}(|E|^{\frac{3}{2}})$ (Dwivedi et al., 2022), with further improvements possible (Fowlkes et al., 2004). Beacon distances can be computed with a Breadth-First-Search (BFS) starting from every beacon node, which is in $\mathcal{O}(|E|)$ as the number of beacons is constant. In practice, we can efficiently compute distances on the GPU using the Bellman-Ford (Bellman, 1958; Ford Jr, 1956) algorithm with unit edge lengths. Random features only require an overhead of $\mathcal{O}(|V|)$. The coarsening can be computed in sub-quadratic time (Jin et al., 2020). K-Nearest-Neighbor graphs can be computed in $\mathcal{O}(|V|\log|V|)$ using K-d-trees and only add a number of edges linear in the number of nodes for fixed K. We thus conclude that CoRe-GD runs in sub-quadratic time overall both for the hierarchical and non-hierarchical approach when using our instantiation.

Table 1: Comparison of scale-invariant stress between classical Graph Drawing methods, learned models, and our proposed framework. Bold numbers mark the best result, underlined numbers the second best. We train all models on the individual datasets for each reported score, except CoRe-GD-mix, which was trained on a mix of all datasets. Our model achieves or matches state-of-the-art on all datasets, including Rome, a popular benchmark in Graph Drawing.

| Model | Rome | ZINC | MNIST | CIFAR10 | PATTERN | CLUSTER |
|---|---|---|---|---|---|---|
| PivotMDS | $388.77 \pm 1.02$ | $29.85 \pm 0.00$ | $173.32 \pm 0.12$ | $384.64 \pm 0.20$ | $3813.30 \pm 1.89$ | $3538.25 \pm 0.73$ |
| neato | $244.22 \pm 0.55$ | $5.76 \pm 0.05$ | $129.87 \pm 0.19$ | $263.44 \pm 0.20$ | $3188.34 \pm 0.59$ | $2920.30 \pm 0.75$ |
| sfdp | $296.05 \pm 1.16$ | $20.02 \pm 0.27$ | $172.63 \pm 0.11$ | $377.97 \pm 0.11$ | $3219.23 \pm 1.31$ | $2952.95 \pm 1.81$ |
| $(sgd)^2$ | $\underline{233.49 \pm 0.22}$ | $\underline{5.14 \pm 0.01}$ | $\underline{129.19 \pm 0.00}$ | $\mathbf{262.52 \pm 0.00}$ | $3181.51 \pm 0.05$ | $2920.36 \pm 0.78$ |
| $(DNN)^2$ | $248.56 \pm 3.01$ | $9.61 \pm 1.62$ | $147.70 \pm 6.93$ | $264.52 \pm 0.32$ | $3100.07 \pm 4.59$ | $2862.95 \pm 16.17$ |
| DeepGD | $235.22 \pm 0.71$ | $6.19 \pm 0.07$ | $129.23 \pm 0.03$ | $262.91 \pm 0.13$ | $3080.70 \pm 0.24$ | $2838.13 \pm 0.08$ |
| CoRe-GD (ours) | $\mathbf{233.17 \pm 0.13}$ | $\mathbf{5.11 \pm 0.02}$ | $\mathbf{129.10 \pm 0.02}$ | $\underline{262.68 \pm 0.08}$ | $3067.02 \pm 0.79$ | $\mathbf{2827.81 \pm 0.36}$ |
| CoRe-GD-mix (ours) | $234.60 \pm 0.10$ | $5.21 \pm 0.02$ | $129.24 \pm 0.01$ | $262.90 \pm 0.02$ | $\mathbf{3066.31 \pm 0.20}$ | $\underline{2828.13 \pm 0.12}$ |

## 4 EMPIRICAL EVALUATION

Our empirical study is split in two: We first demonstrate the effectiveness of the CoRe-GD framework on the Rome dataset, a common benchmark in the Graph Drawing community. We further add more datasets of similar sizes, as these allow us to train and evaluate all models but also let us assess their performance on different graph distributions. As these graphs are rather small ($\sim 100$ nodes), we do not need to employ the coarsening yet. Instead, we can focus on the improvements that can be made on a single level. For a study of scalability and the application of CoRe-GD to larger graphs, we later use a subset of graphs from the suitesparse matrix collection and generate random graphs to compare the runtime and quality of layouts. Code for CoRe-GD is available online [1]. For more background information on benchmarking for Graph Drawing tasks, we refer to Appendix A.7.4.

To quantify the quality of drawings for a dataset $D$, we report two metrics: The mean scale-invariant stress over all graph instances and the mean of scale-invariant stresses normalized by the number of node pairs in the graph:

$$\frac{1}{|D|} \sum_{G \in D} \text{stress}(G, \alpha_{G,\Gamma_G} \cdot \Gamma_G), \qquad \text{and} \qquad \frac{1}{|D|} \sum_{G=(V,E) \in D} |V|^{-2} \text{stress}(G, \alpha_{G,\Gamma_G} \cdot \Gamma_G).$$

We report both the normalized and unnormalized stress as bigger graphs usually incur higher stress, which distorts the mean for mixed-size datasets (Giovannangeli et al., 2021; Tiezzi et al., 2022). Standard deviation is reported over 5 different random seeds in all runs. Our baseline for CoRe-GD uses random, distance beacons and laplacian PEs as well as KNN rewiring. We further investigate these choices with an ablation study in Appendix A.4, where we test different convolutions, rewiring methods, and initial features. Notably, the positional rewiring method improves performance. Further, we also train models without latent embeddings where only the positions are passed between layers and observe considerably worse performance. An application of the latent node embeddings generated by CoRe-GD as positional encodings for graph transformers is investigated as a downstream task in Appendix A.6.

**Baseline comparison.** We compare the performance of CoRe-GD on the Rome dataset, a popular benchmark in the Graph Drawing community that consists of graphs curated by the University of Rome (Di Battista et al., 1997) and use the same data split as introduced by Wang et al. (2021). We further extend the evaluation by including datasets from different application domains such as molecular graphs, superpixel graphs for images, and graphs that were generated with a stochastic block model that we take from Dwivedi et al. (2022) (originally proposed for GNN benchmarking). This new benchmarking framework allows us to test the performance of all algorithms on a diverse set of tasks. We use the same hyperparameter configuration that achieved the best performance on Rome for all other datasets.

PivotMDS (Brandes & Pich, 2007), neato (Kamada & Kawai, 1989), sfdp (Hu, 2005) and $(sgd)^2$ (Ahmed et al., 2022) are included in the comparison as classical, non-learning-based drawing techniques. For learning-based approaches, we compare to DeepGD (Wang et al., 2021) and

---

[1] https://github.com/floriangroetschla/CoRe-GD

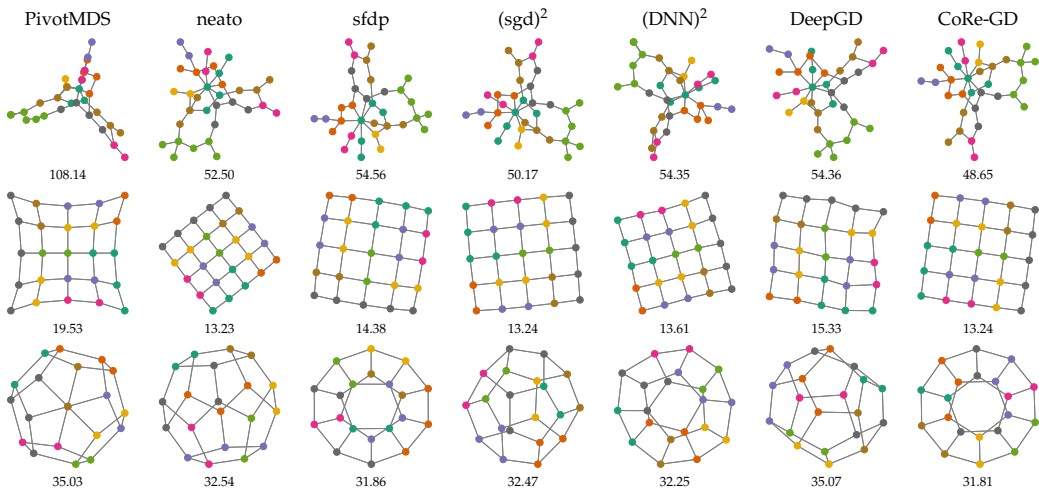

Figure 4: Visual comparison of Graph Drawings between classical algorithms, learned models, and CoRe-GD. Below each drawing, we report the scale-invariant stress. All neural models were trained on the Rome dataset. CoRe-GD generates good layouts across the board, even for regular graphs.

Table 2: Comparison on a subset of graphs from the suitesparse matrix collection. We note that CoRe-GD maintains competitive performance with classical baselines and underscore the necessity of coarsening in achieving such results.

| | **CoRe-GD** | | | | | |
| Metric | coarsening | no coarsening | PivotMDS | neato | sfdp | $(sgd)^2$ |
| --- | --- | --- | --- | --- | --- | --- |
| stress | $42639 \pm 313$ | $45573 \pm 844$ | $65404 \pm 907$ | $43110 \pm 274$ | $48386 \pm 321$ | $42435 \pm 149$ |

$(DNN)^2$ (Giovannangeli et al., 2021). Further training details and dataset information can be found in Appendix A.7.4. Table 1 summarizes the results. CoRe-GD, $(sgd)^2$ and DeepGD are trained on each respective dataset, while CoRe-GD-mix was trained on a combination of all datasets by taking the first 10k graphs from each training set. CoRe-GD outperforms all other methods on all datasets except CIFAR10, where it is slightly outperformed by $(sgd)^2$, the best-performing classical algorithm. Even though DeepGD uses message-passing on the fully connected graph with pre-computed pairwise node distances, it always performs worse than CoRe-GD. CoRe-GD-mix performs well overall, albeit a bit worse than CoRe-GD trained on the respective datasets, except for PATTERN, where it outperforms all other models. This demonstrates that while CoRe-GD can beat other models on data trained from the same distribution, it is also possible to train a general-purpose version of CoRe-GD that can be used for a wide variety of graphs from different distributions. A table reporting the normalized stress values can be found in Appendix A.7.2. We do not observe any major differences between normalized and non-normalized stress. For a qualitative comparison, Figure 4 depicts graph samples drawn with all methods. The differences become most apparent for the grid graph, where it is optimal to position nodes as a rectangular grid. CoRe-GD approximates this well, while especially PivotMDS and DeepGD generate distorted drawings.

**Scalability.** To demonstrate that CoRe-GD can scale beyond small graphs, we create a new dataset by sampling graphs from the suitesparse matrix collection. These graphs represent real-world instances stemming from fields such as power network problems, structural problems, circuit simulation, and more. We apply hierarchical coarsening and investigate its impact on CoRe-GD. We were not able to train DeepGD due to memory limitations on the GPU, as message passing on the fully connected graph leads to too many gradients that accumulate on the device. Similarly, while already lagging behind on the smaller datasets, we did not manage to get the training of $(DNN)^2$ to converge. Results are shown in Table 2. We can observe that CoRe-GD performs competitively to existing baselines and that the coarsening is crucial to achieving that performance.

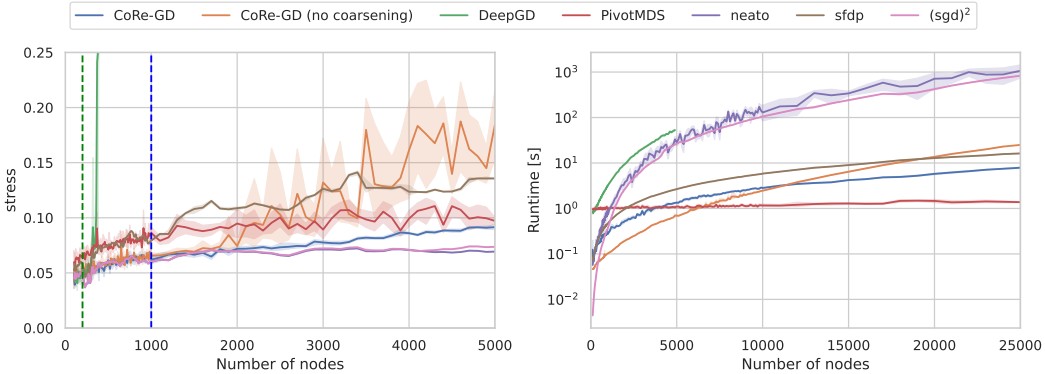

Figure 5: Normalized stress and runtimes on randomly generated sparse graphs (Delaunay triangulations of random point clouds). The green vertical line at 200 marks the train limit of DeepGD, while the blue line at 1,000 marks the limit of CoRe-GD. DeepGD could only be executed for graphs up to size 5,000 before running out of GPU memory. DeepGD becomes unstable under out-of-distribution data, whereas CoRe-GD slightly degrades but still keeps up with the other baselines. The runtimes show that CoRe-GD scales better than most baselines. Further, coarsening aids with scalability for larger graphs, as the initial feature computation, which becomes progressively more expensive, only has to be done for the coarsest graph. See Appendix A.7.3 for more details on the used setup.

We further test the scaling behavior of the models when presented with sparse graphs from the same distribution but of increasing sizes. We train CoRe-GD on sparse Delaunay triangulations of random point clouds up to size 1,000. DeepGD was trained on graphs up to size 200, as bigger graphs result in the aforementioned memory problems. This also happened in inference when presented with graphs of size 5,000 or bigger, due to the all-pairs-shortest-path information that has to be computed and kept in memory. In Figure 5 (left), we can observe that the training size is clearly visible for DeepGD, which does not generalize well beyond its training distribution. CoRe-GD without coarsening suffers to a lesser degree, whereas CoRe-GD with coarsening manages to compete with the other baselines, albeit degrading for graphs beyond its training distribution. Computing the stress of a layout is slow and memory-intensive, which is why we cap the analysis at 5,000 nodes. As inference can be done without stress computation, we measure the real-world runtime of all models on graphs up to size 25,000. We notice that, initially, CoRe-GD performs faster without coarsening compared to its coarsened counterpart. However, as the graph size increases, the computation of initial features scales with the graph size in the absence of coarsening. In contrast, when coarsening is employed, this initial feature computation is only required for the coarsest, smaller graph. Overall, CoRe-GD scales well when compared to other baselines, with PivotMDS being the clear exception.

## 5 CONCLUSION

We introduce CoRe-GD, a neural framework for Graph Drawing. Our approach incorporates coarsening and a novel positional rewiring technique to achieve scalability and maintain computational efficiency. To judge the performance of CoRe-GD, we conducted an evaluation encompassing established benchmark datasets such as the Rome dataset, datasets from the graph learning community, and real-world graphs sourced from the suitesparse matrix collection. The results demonstrate that CoRe-GD consistently delivers good performance and extends its capabilities to graph sizes not encountered during training. CoRe-GD's generation of latent node embeddings, rather than solely computing positions, expands its potential applications. Notably, CoRe-GD also stands out as the first neural architecture capable of achieving this level of scalability, otherwise only known from sophisticated handcrafted algorithms. To accomplish this objective, we introduced encoded beacon distances as node features and employed a replay buffer to store latent embeddings for training deep recurrent GNNs, which lets us run hundreds of graph convolutions recurrently to improve the layout without destabilizing. Both methods might hold the potential for addressing a broader range of geometric graph-related problems beyond traditional graph visualization.

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

# A  APPENDIX

## A.1  BASELINE ALGORITHMS

Table 3: Comparison of classical and learned graph drawing algorithms. Our method can facilitate information exchange in the graph while maintaining an efficient runtime and achieving state-of-the-art on the Rome dataset. For $(sgd)^2$, a sparse approximation with sub-quadratic runtime exists. (Ahmed et al., 2022)

| Model | Learned | Sub-Quadratic Runtime | Global Message Passing | Mean Stress on Rome Dataset↓ |
|---|---|---|---|---|
| PivotMDS (Brandes & Pich, 2007) | × | ✓ | N/A | 388.77 |
| neato (Kamada & Kawai, 1989) | × | × | N/A | 244.22 |
| sfdp (Hu, 2005) | × | ✓ | N/A | 296.05 |
| $(sgd)^2$ (Ahmed et al., 2022) | × | (×) | N/A | 233.49 |
| $(DNN)^2$ (Giovannangeli et al., 2021) | ✓ | ✓ | × | 248.56 |
| DeepGD (Wang et al., 2021) | ✓ | × | ✓ | 235.22 |
| CoRe-GD (ours) | ✓ | ✓ | ✓ | **233.17** |

An overview of the used baselines and their complexities is given in Table 3. CoRe-GD is the only algorithm that (1) is learnable, i.e., can be adapted to different graph distributions by training on them, (2) has sub-quadratic runtime complexity, thus scaling well when applied to bigger graphs, and (3) allows for global message exchange by alleviating potential information bottlenecks with positional rewiring. Notably, CoRe-GD outperforms prior work on the Rome dataset and generally performs better than all learned baselines.

## A.2  ARCHITECTURE DETAILS

---

**Algorithm 1** CoRe-GD

---

1: **procedure** LAYOUT OPTIMIZATION($\mathcal{G}_l = (V_l, E_l), h_1^l, r$)  ▷ Figure 2 top
2:    **for** $i = 1$ to $r$ **do**  ▷ Optimize layout for $r$ rounds
3:        $h_{i*}^l \leftarrow \text{Conv}_\text{E}(E_l, h_1^l)$  ▷ Convolution on $\mathcal{G}_l$
4:        $\Gamma_i^l \leftarrow \text{Dec}(h_{i*}^l)$  ▷ Decode node positions
5:        $E_R \leftarrow$ Rewiring based on $\Gamma_i^l$
6:        $h_{i+1}^l \leftarrow \text{Conv}_\text{R}(E_R, h_{i*}^l)$
7:    **end for**
8:    $h_{r+1}^{l*} \leftarrow \text{Conv}_\text{E}(E_l, h_{r+1}^l)$  ▷ Finish with convolution on original topology
9:    **return** $h_{r+1}^{l*}$
10: **end procedure**

11: **procedure** CORE-GD($\mathcal{G}$)  ▷ Figure 2 bottom
12:    $\mathcal{G}_1 = (V_1, E_1), \ldots, \mathcal{G}_c = (V_c, E_c) \leftarrow$ Coarsen $\mathcal{G}$ into hierarchy  ▷ $\mathcal{G}_c = \mathcal{G}$
13:    $h_1^1 \leftarrow$ Compute and encode initial features for $\mathcal{G}_1$
14:    **for** $l = 1$ to $c - 1$ **do**  ▷ Iterate through coarsening levels $l$
15:        $h_{r+1}^l \leftarrow$ LAYOUT OPTIMIZATION($\mathcal{G}_l, h_1^l, r$)
16:        $h_1^{l+1} \leftarrow$ Project embeddings $h_{r+1}^{l*}$ to $\mathcal{G}_{l+1}$ and add noise
17:    **end for**
18:    $h_{r+1}^{c+1} \leftarrow$ LAYOUT OPTIMIZATION($\mathcal{G}_c, h_1^{c+1}, r$)
19:    $\Gamma_{r+1}^{c+1} \leftarrow \text{Dec}(h_{r+1}^{c+1})$
20:    **return** $\Gamma_{r+1}^{c+1}$  ▷ Return final positions
21: **end procedure**

---

Algorithm 1 provides further details on the modules depicted in Figure 2, with the same notation for intermediate states $h$. The projection of embeddings is done with a sparse matrix multiplication and first sets embeddings for all nodes to the embedding of their supernode. As this means that

multiple nodes can receive the same embeddings, we add noise from a normal distribution to change them slightly. It should be noted that CoRe-GD only decodes latent embeddings to positions for the purpose of computing the rewiring or to get the final output of the network. In all other steps, we pass the latent embeddings to let nodes maintain more information than just their current position.

---

**Algorithm 2** CoRe-GD Training

1: **procedure** CORE-GD TRAIN(Batch $\mathcal{B} = ((\mathcal{G}_1, h_1), \ldots, (\mathcal{G}_B, h_B)$, replay buffer $\mathcal{R}$)
2:    **if** $p_{\text{uncoarsen}} \geq \mathbf{X} \sim U(0, 1)$ **then**                    ▷ Uncoarsen graphs with probability $p_{\text{uncoarsen}}$
3:        $r_{\text{pre}} \leftarrow \mathcal{N}(0, \sigma_{\text{pre}})$
4:        $r_{\text{post}} \leftarrow \mathcal{N}(0, \sigma_{\text{post}})$
5:        **for** $(\mathcal{G}, h)$ in $\mathcal{B}$ **do**                              ▷ in parallel
6:            $h_{\text{new}} \leftarrow$ LAYOUT OPTIMIZATION$(\mathcal{G}, h, r_{\text{pre}})$
7:            **if** $\mathcal{G}$ can be uncoarsened **then**
8:                $\mathcal{G}_{\text{new}} \leftarrow$ Next graph in the coarsening hierarchy
9:                $h_{\text{new}} \leftarrow$ Project embeddings $h_{\text{new}}$ to $\mathcal{G}_{\text{new}}$ and add noise
10:            **end if**
11:            $h_{\text{new}} \leftarrow$ LAYOUT OPTIMIZATION$(\mathcal{G}_{\text{new}}, h_{\text{new}}, r_{\text{post}})$
12:        **end for**
13:    **else**d
14:        $r \leftarrow \mathcal{N}(0, \sigma)$
15:        **for** $(\mathcal{G}, h)$ in $\mathcal{B}$ **do**                              ▷ in parallel
16:            $h_{\text{new}} \leftarrow$ LAYOUT OPTIMIZATION$(\mathcal{G}, h, r_{\text{post}})$
17:            $\mathcal{G}_{\text{new}} \leftarrow \mathcal{G}$
18:        **end for**
19:    **end if**
20:    **for** $(\mathcal{G}, h_{\text{new}})$ in $\mathcal{B}$ **do**
21:        $\Gamma \leftarrow \text{Dec}(h)$
22:        Compute and collect loss for layout $\Gamma$
23:        Randomly replace elements in $\mathcal{R}$ with $(\mathcal{G}, h_{\text{new}})$
24:    **end for**
25:    Do backpropagation on collected losses
26: **end procedure**

---

Training and backpropagating through all layers at once is costly (in terms of memory) and tends to become unstable. Still, we want the trained network to be able to run hundreds of the same parametrized GNN-convolutions during inference to optimize the layout of a big graph. We achieve this goal by training only with few rounds of the layout optimization, but storing the resulting latent embeddings together with the graph in a replay buffer $\mathcal{R}$ such that they can be used in a future training batch. Doing so, we only backpropagate through a small (randomized) number of layers at a time. Pseudocode for the training of a batch is provided in Algorithm 2. Batches are drawn alternating from replay buffer $\mathcal{R}$ and original graphs with the encoder Enc applied before being passed to the procedure.

### A.2.1    GRAPH CONVOLUTION

We use an adapted version of the GIN and GRU convolution. Assuming that $h_v^t$ is the embedding of node $v$ at step $t$, the GRU convolution looks as follows:

$$h_v^{t+1} = \text{GRU}\left(\left(\sum_{w \in N(v)} \Theta(h_v^t \| h_w^t)\right), h_v^t\right),$$

where GRU is a Gated Recurrent Unit and, $\Theta$ a learnable MLP and $\|$ the concatenation operation. We can observe that instead of only aggregating node features from neighbors, they are first combined with the embedding of the target node that receives the message. We do the same adaption for the GIN convolution:

$$h_v^{t+1} = \Theta_1\left((1+\epsilon) \cdot h_v^t + \sum_{w \in N(v)} \Theta_2(h_v^t \| h_w^t)\right).$$

The GAT convolution is not adapted that way, as it already incorporates this relative information. For the GNN convolution on the input graph topology, we use two layers of the same convolution, i.e., do two rounds of message passing before interleaving it with one round of message-passing on the rewired topology.

### A.2.2 LAPLACIAN PE COMPUTATION

We apply the same Laplacian eigenvector positional encodings as Dwivedi et al. (2022), via the factorization of the graph Laplacian:

$$\Delta = I - D^{-1/2}AD^{-1/2} = U^T\Lambda U,$$

with $A$ being the adjacency matrix of the graph, $D$ the degree matrix and $U$ and $\Lambda$ the eigenvectors and eigenvalues of the eigen-decomposition. We always take the smallest Laplacian eigenvectors, denoted in 7 as "Number of Laplacian eigenvectors".

### A.3 REWIRING

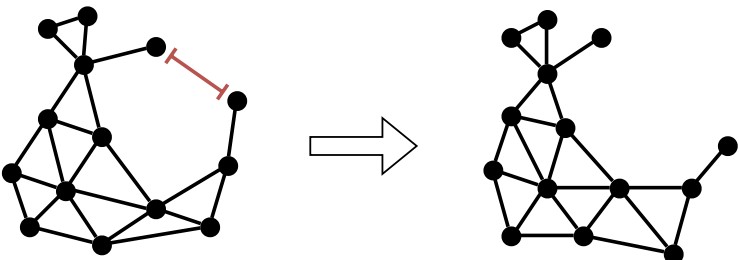

Figure 6: Motivation for the positional rewiring. Two nodes that are far apart in the graph are positioned close together in the drawing. To let them exchange information directly, we rewire the graph based on intermediate node positions. After the layout is optimized, the nodes are moved further apart from each other.

Our positional rewiring technique is based on decoding intermediate embeddings to get current positions. Figure 6 motivates the rewiring further. Here, two nodes are positioned close to each other in the current drawing, while they should have a distance proportional to their shortest path length, which is large in this case. The two nodes are not able to exchange information directly with each other, and message-passing would take many rounds to propagate the positioning sufficiently. The positional rewiring counters this by directly connecting these nodes and letting them optimize their relative positioning.

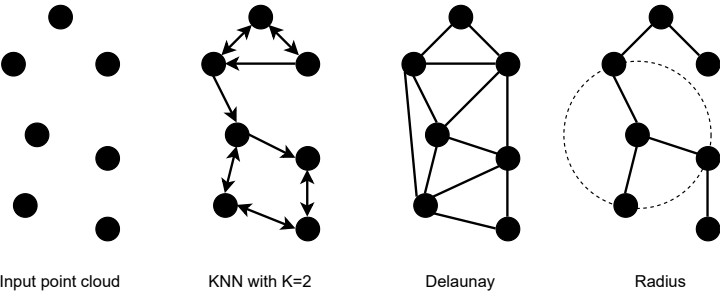

Figure 7: The same point cloud rewired with three techniques.

Without considering the underlying graph topology, one can consider these positions $\Gamma(V)$ as a point cloud in a $n$-dimensional space. The point clouds are then converted into a graph for localized message passing. We experimented with three rewiring methods: K-nearest-neighbor graphs, Delaunay triangulations, and radius graphs. Figure 7 shows the same point cloud, rewired with the three methods.

Table 4: Ablation study on CoRe-GD. Each row reports the scale-invariant stress corresponding to a single change of the architecture or training configuration. All runs were done on the Rome dataset.

| | Ablation | Stress | Normalized Stress $\times 10^3$ |
|---|---|---|---|
| | Baseline | $233.17 \pm 0.13$ | $57.61 \pm 0.03$ |
| Conv | GIN conv | $238.53 \pm 1.43$ | $59.17 \pm 0.41$ |
| | GAT conv | $270.83 \pm 8.45$ | $70.27 \pm 2.10$ |
| Initial features | No beacons | $233.39 \pm 0.30$ | $57.65 \pm 0.07$ |
| | No laplacian PE | $234.65 \pm 0.33$ | $57.91 \pm 0.08$ |
| | No random features | $233.24 \pm 0.17$ | $57.64 \pm 0.05$ |
| Rewiring | No rewiring | $236.38 \pm 0.12$ | $58.50 \pm 0.02$ |
| | Delaunay | $233.40 \pm 0.67$ | $57.69 \pm 0.19$ |
| | Radius | $234.14 \pm 0.68$ | $57.81 \pm 0.17$ |
| | Random | $236.01 \pm 0.08$ | $58.38 \pm 0.03$ |
| | No replay buffer | $236.65 \pm 1.75$ | $58.61 \pm 0.50$ |
| | Scale-dependent stress | $234.04 \pm 0.82$ | $57.83 \pm 0.21$ |

Table 5: Ablation study for different coarsening approaches, taken from Jin et al. (2020), on the suitesparse dataset.

| Metric | heavy_edge | affinity_GS | kron | variation_neighborhoods |
|---|---|---|---|---|
| stress | $42800 \pm 422$ | $42639 \pm 313$ | $42699 \pm 248$ | $42935 \pm 319$ |
| normalized stress ($\times 10^3$) | $96.7 \pm 0.6$ | $96.4 \pm 0.7$ | $96.6 \pm 0.3$ | $97.7 \pm 1.5$ |

**KNN** For the KNN graph we add an edge $(u, v)$ for the K closest points $u$ to $v$ regarding Euclidean distance. This means the graph is directed and every node has an in-degree of exactly K. In the message-passing step, messages are only sent along the directed edges, letting every node know about its K nearest neighbors.

**Delaunay Triangulation** The Delaunay triangulation is the dual graph of the Voronoi diagram. In a Delaunay triangulation, it holds that given any triangle, no vertex lies inside its circumcircle. As we have a triangulation, the number of edges is linear in the number of nodes. Moreover, the fast sequential computation time of $\mathcal{O}(|V|\log|V|)$ makes it an ideal fit for our rewiring technique. The downside of this approach is that it is hard to parallelize the computation, making it slow in practice. This is also what we observed during our testing.

**Radius Graph** In the Radius graph, two nodes are connected if the Euclidean distance between them is smaller or equal to a predefined threshold. For the testing in our ablation study, we set this value to $0.05$. Setting a good radius is hard, and choosing a bad value can lead to bad performance as the number of edges is not bounded with this approach, and we can get a fully connected graph in the worst case. The radius can also be sensitive to the graph size we want to draw, especially as we restrict the drawing area to $[0, 1]^d$. Here, bigger graphs will populate the space more densely. For these reasons we prefer the KNN or Delaunay based rewiring over the Radius graph.

## A.4 Ablation Study

We corroborate our architectural choices with an ablation study on the Rome dataset. As can be deduced from Table 4, every component contributes to the performance of CoRe-GD. Most notably, the chosen convolution plays a major role, while the chosen rewiring method and the usage of the replay buffer also make a significant impact. This supports the claim that the GRU convolution is well-suited for recurrent GNNs. While the Delaunay rewiring performed similarly to KNN, training time is much longer for Delaunay as the triangulation is computed on the CPU. A GPU-optimized implementation could help to make it more viable in practice.

### A.5 SCALE-INVARIANT STRESS

Restricting the domain of the graph layout can limit the solution space to become suboptimal. Therefore, we want to rescale node positions using a scaling factor $\alpha$ for node positions $P$ such that the scaled layout $P_\alpha := \{\alpha \cdot p \mid p \in P\}$ has optimal stress among all $\alpha$. Therefore, the stress we report throughout the paper becomes *scale-invariant*. Recall the definition of stress:

$$\text{stress}(G, \Gamma) := \sum_{u,v \in V, u \neq v} w_{uv}(\|\Gamma(u) - \Gamma(v)\|_2 - d_{uv})^2$$

We want to find the optimal $\alpha_{G,\Gamma}$ which minimizes the stress loss:

$$\alpha_{G,\Gamma} = \arg\min_{\alpha \in \mathbb{R}} \text{stress}(G, \Gamma, \alpha) = \arg\min_{\alpha \in \mathbb{R}} \sum_{u,v \in V, u \neq v} w_{uv}(\|\alpha\Gamma(u) - \alpha\Gamma(v)\|_2 - d_{uv})^2$$

To find the optimal $\alpha$, we take the first derivative with respect to $\alpha$ and solve it to be 0.

$$0 \overset{!}{=} \frac{\partial}{\partial \alpha} \text{stress}(G, \Gamma, \alpha)$$

$$0 = \frac{\partial}{\partial \alpha} \sum_{u,v \in V, u \neq v} w_{uv}(\|\alpha\Gamma(u) - \alpha\Gamma(v)\|_2 - d_{uv})^2$$

$$0 = \frac{\partial}{\partial \alpha} \sum_{u,v \in V, u \neq v} w_{uv}(\alpha\|\Gamma(u) - \Gamma(v)\|_2 - d_{uv})^2$$

$$0 = \sum_{u,v \in V, u \neq v} w_{uv}(2\alpha\|\Gamma(u) - \Gamma(v)\|_2^2 - 2\|\Gamma(u) - \Gamma(v)\|_2 d_{uv})$$

$$\sum_{u,v \in V, u \neq v} w_{uv} 2\|\Gamma(u) - \Gamma(v)\|_2 d_{uv} = \sum_{u,v \in V, u \neq v} w_{uv} 2\alpha\|\Gamma(u) - \Gamma(v)\|_2^2$$

$$\sum_{u,v \in V, u \neq v} w_{uv}\|\Gamma(u) - \Gamma(v)\|_2 d_{uv} = \alpha \sum_{u,v \in V, u \neq v} w_{uv}\|\Gamma(u) - \Gamma(v)\|_2^2$$

$$\alpha = \frac{\sum_{u,v \in V, u \neq v} w_{uv}\|\Gamma(u) - \Gamma(v)\|_2 d_{uv}}{\sum_{u,v \in V, u \neq v} w_{uv}\|\Gamma(u) - \Gamma(v)\|_2^2}$$

Moreover, we can verify that the solution is the unique minimum by checking if the second derivative with respect to $\alpha$ is positive.

$$\frac{\partial^2}{\partial^2 \alpha} \sum_{u,v \in V, u \neq v} w_{uv}(\alpha\|\Gamma(u) - \Gamma(v)\|_2 - d_{uv})^2 = \sum_{u,v \in V, u \neq v} w_{uv}(2\|\Gamma(u) - \Gamma(v)\|_2^2)^2 > 0$$

Therefore, the following closed form solution defines the optimal $\alpha_{G,\Gamma}$:

$$\alpha_{G,\Gamma} = \arg\min_{\alpha \in \mathbb{R}} \text{stress}(G, \Gamma, \alpha) = \frac{\sum_{u,v \in V, u \neq v} w_{uv}\|\Gamma(u) - \Gamma(v)\|_2 d_{uv}}{\sum_{u,v \in V, u \neq v} w_{uv}\|\Gamma(u) - \Gamma(v)\|_2^2}$$

### A.6 CORE-GD AS POSITIONAL ENCODING

By utilizing stress as the loss, we can train CoRe-GD in an unsupervised fashion to generate node embeddings that capture pair-wise node distances in the graph. In the case of graph drawing, we usually use latent embeddings to find a good layout in a low-dimensional space. However, these latent embeddings capture inherent structural information of the graph, which might be useful for other learning applications, e.g., in improving the performance of downstream tasks. We test the downstream performance using the GPS framework (Rampášek et al., 2022), where we run baselines using initial features consisting of laplacian eigenvectors, random-walk structural encodings (RWSE), and beacon encodings to test CoRe-GD's latent embeddings as well as the derived positions. As seen in Table 6, CoRe-GD embeddings admit competitive performance, outperforming

Table 6: Performance using the GPS framework using different types of positional embeddings. Models marked with a * were trained by us.

| Model | ZINC ↓ | MNIST ↑ | CIFAR10 ↑ | PATTERN ↑ | CLUSTER ↑ |
|---|---|---|---|---|---|
| DGN (Beaini et al., 2021) | $0.168 \pm 0.003$ | - | $\mathbf{72.838 \pm 0.417}$ | $86.680 \pm 0.034$ | - |
| CIN (Bodnar et al., 2021) | $\underline{0.079 \pm 0.006}$ | - | - | - | - |
| GIN-AK+ (Zhao et al., 2021) | $\underline{0.080 \pm 0.001}$ | - | $72.19 \pm 0.13$ | $\mathbf{86.850 \pm 0.057}$ | |
| K-Subgraph SAT (Chen et al., 2022) | $0.094 \pm 0.008$ | - | - | $\underline{86.848 \pm 0.037}$ | $77.856 \pm 0.104$ |
| EGT (Hussain et al., 2022) | $0.108 \pm 0.009$ | $\mathbf{98.173 \pm 0.087}$ | $68.702 \pm 0.409$ | $86.821 \pm 0.020$ | $\mathbf{79.232 \pm 0.348}$ |
| GPS+LapPE (Rampášek et al., 2022) | $0.116 \pm 0.009$ | $\underline{98.051 \pm 0.126}$ | $72.298 \pm 0.356$ | $86.685 \pm 0.059$ | $78.016 \pm 0.180$ |
| GPS+RWSE (Rampášek et al., 2022) | $\mathbf{0.070 \pm 0.004}$ | - | $71.958 \pm 0.398$ | - | - |
| GPS+CoRe-GD* | $0.126 \pm 0.010$ | $97.876 \pm 0.140$ | $72.494 \pm 0.340$ | $86.522 \pm 0.405$ | $77.454 \pm 0.157$ |
| GPS+CoRe-GD* | $0.089 \pm 0.006$ | $97.970 \pm 0.190$ | $71.614 \pm 0.608$ | $86.769 \pm 0.045$ | $\underline{78.316 \pm 0.169}$ |
| GPS+CoRe-GD-pos* | $0.122 \pm 0.009$ | $97.884 \pm 0.154$ | $\underline{72.296 \pm 0.286}$ | $86.721 \pm 0.020$ | $77.722 \pm 0.168$ |

Table 7: Final hyperparameter settings and tested configurations for CoRe-GD.

| Hyperparameter | Baselines | Tested for baselines | suitesparse | Delaunay graphs |
|---|---|---|---|---|
| Hidden dimension | 64 | {32, 64, 128} | 64 | 64 |
| Dropout | 0.0 | {0.0, 0.1} | 0.0 | 0.0 |
| Mean number of rounds | 5 | {4, 5, 6} | 5 | 5 |
| Variance of round number | 1 | {1} | 1 | 1 |
| Batch size | 16/32 | {16, 32, 64} | 4 | 4 |
| Convolution | GRU | {GIN, GAT, GRU} | GRU | GRU |
| # Laplacian eigenvectors | 8 | {4, 6, 8} | 8 | 8 |
| # Random inputs | 1 | {0, 1} | 1 | 1 |
| # Beacons | 2 | {1, 2, 4, 8} | 2 | 2 |
| Encoding size per beacon | 8 | {2, 4, 8, 16} | 8 | 8 |
| Rewiring | KNN | {KNN, Delaunay, Radius} | KNN | KNN |
| K for KNN | 8 | {4, 6, 8} | 8 | 8 |
| Replay buffer size | 4096 | {2048, 4096} | 1024 | 1024 |
| Coarsen algorithm | - | - | affinity_GS | heavy_edge |
| Coarsen reduction factor | - | - | 0.8 | 0.8 |

laplacian and beacon embeddings on ZINC, PATTERN, and CLUSTER, which indicates that CoRe-GD adds additional value to these embeddings through unsupervised stress minimization. Latent embeddings perform better than positions overall, indicating that they contain additional structural information. This hints at the potential of CoRe-GD embeddings for downstream tasks.

## A.7 EXPERIMENTAL SETUP

### A.7.1 TRAINING DETAILS

The final version of CoRe-GD was trained with the same hyperparameters (except batch size) on all datasets. Table 7 lists the setup and tested hyperparameters for the final model we conducted our experiments with. Due to computational constraints, a full grid search of all hyperparameters was infeasible. Therefore, we tuned by running a line search amongst one of the dimensions. We used the Adam optimizer with an initial learning rate of 0.0002 scheduled with the Plateau technique and patience of 12, threshold of 2, and factor of 0.7. Model selection was done by choosing the epoch with the best validation score, and we trained for 200 epochs in total. Elements in the replay buffer were replaced with a probability of 50% if the batch was part of the original dataset and a probability of 100% if the batch was sampled from the replay buffer. All training was done on an RTX 3090 with 24GB of VRAM. For all results reported in the paper we use an output dimension of $d = 2$.

For DeepGD, we used the same configuration as provided by the authors, except for a reduced batch size of 16 for PATTERN, CLUSTER, and CIFAR10 due to the increased size of the graphs in the dataset. The values reported in the paper were produced with PivotMDS initializations, and the number of pivots was set to 10. We used the recommended epoch number of around 600, except when we hit the timeout of 48 hours. The model with the best validation loss was selected. All training was done on an A6000 with 48GB of VRAM.

Table 8: Comparison of scale-invariant stress between classical Graph Drawing methods, learned models, and our proposed model CoRe-GD. We train all models on the individual datasets for each reported score, except CoRe-GD-mix which was trained on a mix of all datasets. Our model achieves or matches state-of-the-art on all datasets, including Rome, a popular benchmark in Graph Drawing. The reported score is the scale-invariant normalized stress multiplied by 1000.

| Model | Rome | ZINC | MNIST | CIFAR10 | PATTERN | CLUSTER |
|---|---|---|---|---|---|---|
| PivotMDS | $105.12 \pm 0.13$ | $54.46 \pm 0.00$ | $34.92 \pm 0.02$ | $27.81 \pm 0.01$ | $257.62 \pm 0.09$ | $249.87 \pm 0.11$ |
| neato | $61.31 \pm 0.10$ | $10.39 \pm 0.09$ | $26.26 \pm 0.04$ | $19.04 \pm 0.01$ | $215.61 \pm 0.03$ | $206.33 \pm 0.07$ |
| sfdp | $77.06 \pm 0.30$ | $35.58 \pm 0.49$ | $34.72 \pm 0.02$ | $27.33 \pm 0.01$ | $217.81 \pm 0.10$ | $208.73 \pm 0.15$ |
| $(sgd)^2$ | $58.01 \pm 0.03$ | $9.30 \pm 0.03$ | $26.13 \pm 0.00$ | $18.98 \pm 0.00$ | $215.31 \pm 0.01$ | $206.50 \pm 0.05$ |
| $(DNN)^2$ | $62.71 \pm 0.91$ | $17.06 \pm 3.05$ | $30.06 \pm 1.51$ | $19.12 \pm 0.02$ | $209.58 \pm 0.34$ | $202.33 \pm 1.28$ |
| DeepGD | $58.55 \pm 0.25$ | $11.21 \pm 0.14$ | $26.13 \pm 0.01$ | $19.01 \pm 0.01$ | $208.28 \pm 0.04$ | $200.43 \pm 0.01$ |
| CoRe-GD (ours) | $57.61 \pm 0.03$ | $9.26 \pm 0.02$ | $26.10 \pm 0.00$ | $18.99 \pm 0.01$ | $207.27 \pm 0.04$ | $199.65 \pm 0.02$ |
| CoRe-GD-mix (ours) | $57.95 \pm 0.04$ | $9.42 \pm 0.04$ | $26.13 \pm 0.00$ | $19.01 \pm 0.00$ | $207.26 \pm 0.02$ | $199.71 \pm 0.01$ |

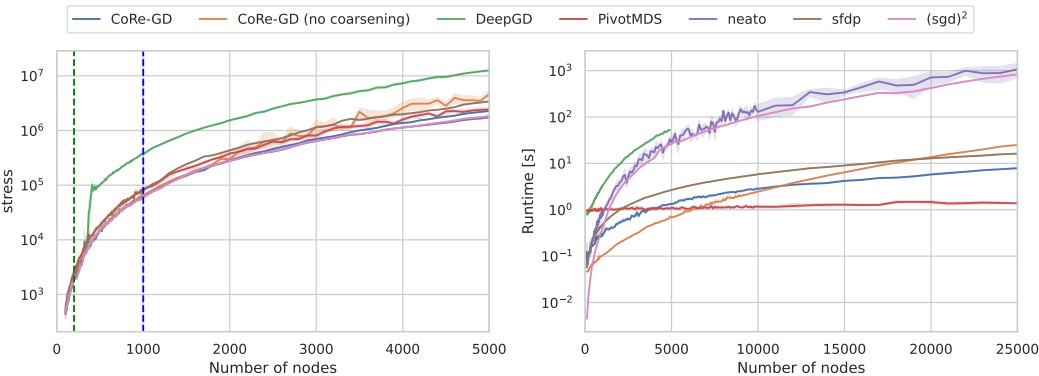

Figure 8: Non-normalized stress for performance and runtimes of the scaling measurements. Otherwise, the same as Figure 5.

For training the $(DNN)^2$ models, we used the default configuration specified by Giovannangeli et al. (2021) for stress optimization. Specifically, we train for at most 200 epochs while using early stopping after 20 epochs if there is no improvement in the validation loss. Furthermore, training time was limited to 48 hours. For each dataset, we trained five models, except for ZINC, which was much more unstable. $(DNN)^2$ on ZINC often did not improve upon the score achieved in the very first epoch. Therefore, we trained 30 models in total to get at least five runs that exhibited a learning behavior and improved after epoch 1. Moreover, a model on the ZINC dataset produced NaN/Inf values in the stress computation for a single graph, which we replaced with a 0 (in favor of $(DNN)^2$). All training was done on CPUs.

### A.7.2 NORMALIZED RESULTS

Table 8 shows normalized results for the Graph Drawing comparison. They are consistent with the non-normalized values, thus further underlining the performance of CoRe-GD.

### A.7.3 SCALING EXPERIMENTS

We run both runtime and performance experiments on 16 physical cores of an AMD EPYC 7742 and 64GB of RAM. For DeepGD and CoRe-GD, we use an RTX 3090 with 24GB VRAM in addition. It should be noted that the preprocessing of DeepGD and the coarsening of CoRe-GD still runs on the CPU, and times are measured, including all data transfers from and to the GPU. For the sake of completeness, we report the non-normalized results for the plot in Figure 8, together with a full view of the DeepGD loss.

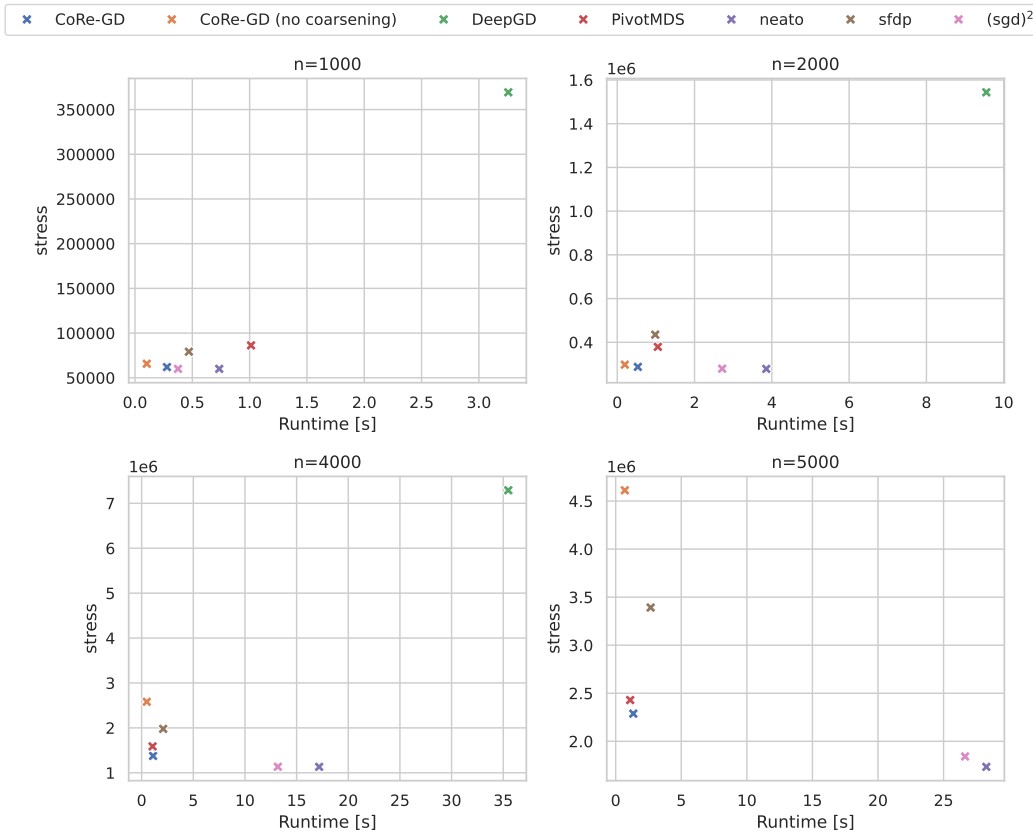

Figure 9: Scatter plots depicting runtime and stress for different graph sizes $n$ as measured for the experiments in Figure 5. CoRe-GD is Pareot-optimal and performs considerably better than DeepGD, which is not part of the last plot as it ran out of memory for 5000 nodes.

To evaluate the tradeoff between achieved loss and runtime further, we plot them together for fixed-size Delaunay graphs in Figure 9. We can observe that CoRe-GD stands as being Pareto-optimal for the four depicted graph sizes.

### A.7.4 GRAPH DRAWING BENCHMARKING AND EMPLOYED DATASETS

Benchmarking for Graph Drawing is highly non-standardized. Classical algorithms like Pivot-MDS (Brandes & Pich, 2007) or $(sgd)^2$ (Ahmed et al., 2022) only use few graphs (some of them usually from the suitesparse matrix collection, others synthetically generated) to evaluate performance comparatively. While this makes it hard to judge the general performance, and the graphs can be sampled with bias, this fails, especially when neural methods are employed that need different sets of graphs for training, validation, and testing. Here, both DNN[2] and DeepGD resort to the Rome dataset by Di Battista et al. (1997) that was introduced for the purpose of experimental comparison of graph drawing algorithms. One downside of this dataset is the fact that it only contains relatively small graphs with 52.3 nodes on average. This makes it unsuitable for testing state-of-the-art algorithms that focus on scalability and larger graph instances. It should be noted that DNN[2] and DeepGD use different random splits of the dataset.

To overcome the limitations of existing benchmarks, we propose the following: We include the Rome dataset with the random split specified by Wang et al. (2021) to compare our results to existing neural methods. As we also want to test the adaptability of the graph drawing algorithm to different graph distributions, we further include the ZINC, MNIST, CIFAR10, PATTERN, and CLUSTER datasets from Dwivedi et al. (2022) that were originally proposed for the purpose of

Table 9: Training dataset statistics for all used benchmarks. Edge counts are for directed edges.

| Dataset | # Graphs | # Nodes | # Edges | License |
|---|---|---|---|---|
| Rome | 11531 | 52.3 | 138.1 | source |
| ZINC | 12,000 | 23.2 | 49.8 | CC-BY 4.0 and MIT |
| MNIST | 55,000 | 70.6 | 564.5 | CC-BY 4.0 and MIT |
| CIFAR10 | 45.000 | 117.6 | 941.2 | CC-BY 4.0 and MIT |
| PATTERN | 10,000 | 118.9 | 6,098.9 | CC-BY 4.0 and MIT |
| CLUSTER | 10,000 | 117.2 | 4,303.9 | CC-BY 4.0 and MIT |
| Suitesparse graphs | 189 | 502.3 | 6385.6 | CC-BY 4.0 |
| Delaunay graphs | 200 | 577.77 | 3428.23 | MIT (self-generated) |

GNN benchmarking. We chose these datasets as they have predefined training/validation/test splits and are roughly the same size as the Rome dataset (with MNIST and CIFAR10 being a bit larger), thus allowing us to train existing neural methods on them. While the original datasets contain labels, we can omit those as training is done in a self-supervised way with the goal of minimizing stress. To further evaluate scalability and performance on larger graphs, we follow (sgd)$^2$ and PivotMDS in taking graphs from the suitesparse matrix collection and creating our own subset with all suitable graphs with 100 to 1000 nodes. For further evaluation of the scaling behavior, we additionally create synthetic Delaunay triangulations. This allows us to create sparse graphs of any size that we can use in the scalability study in Section 4.

We show dataset statistics in Table 9. For any dataset that contains directed graphs, we make them undirected to fit the task. We further only consider connected graphs. Disconnected graphs can always be drawn by drawing each connected component separately.

**Rome (Di Battista et al., 1997)** The Rome dataset is provided by graphdrawing.org and contains graphs collected at the University of Rome. They represent Entity-Relationship diagrams and Data-Flow graphs mainly used for database and software visualization. It is a common benchmark for stress optimization in the graph drawing community.

**Benchmarking GNNs Datasets (Dwivedi et al., 2022)** ZINC contains molecular graphs from the ZINC database of compounds for virtual screening. MNIST and CIFAR10 are generated from the classical MNIST and CIFAR10 datasets by extracting superpixels, small regions of homogeneous intensity in images. PATTERN and CLUSTER are synthetic tasks where graphs were generated with the Stochastic Block Model. Statistics for all datasets are shown in Table 9.

**Suitesparse Subset** We select a subset of suitesparse matrices (https://sparse.tamu.edu/). We consider all quadratic matrices with 100 to 1000 columns that result in an undirected, connected graph. This results in a dataset with graphs from many different distributions and application areas, such as power network problems, structural problems, or circuit simulation problems, to name but a few.

**Random Delaunay Triangulations** We randomly sample the size of the graph between 100 and 1000 and create a random 2D point cloud with that many points. A Delaunay triangulation is computed on the point cloud, resulting in a graph with a planar embedding. The planarity guarantees that the graph has a linear number of edges, keeping it sparse.

## A.8 MORE COARSENING EXAMPLES

Figure 10 shows additional examples of the progressive improvement and uncontraction of CoRe-GD drawings. The graphs are taken from the suitesparse matrix collection.

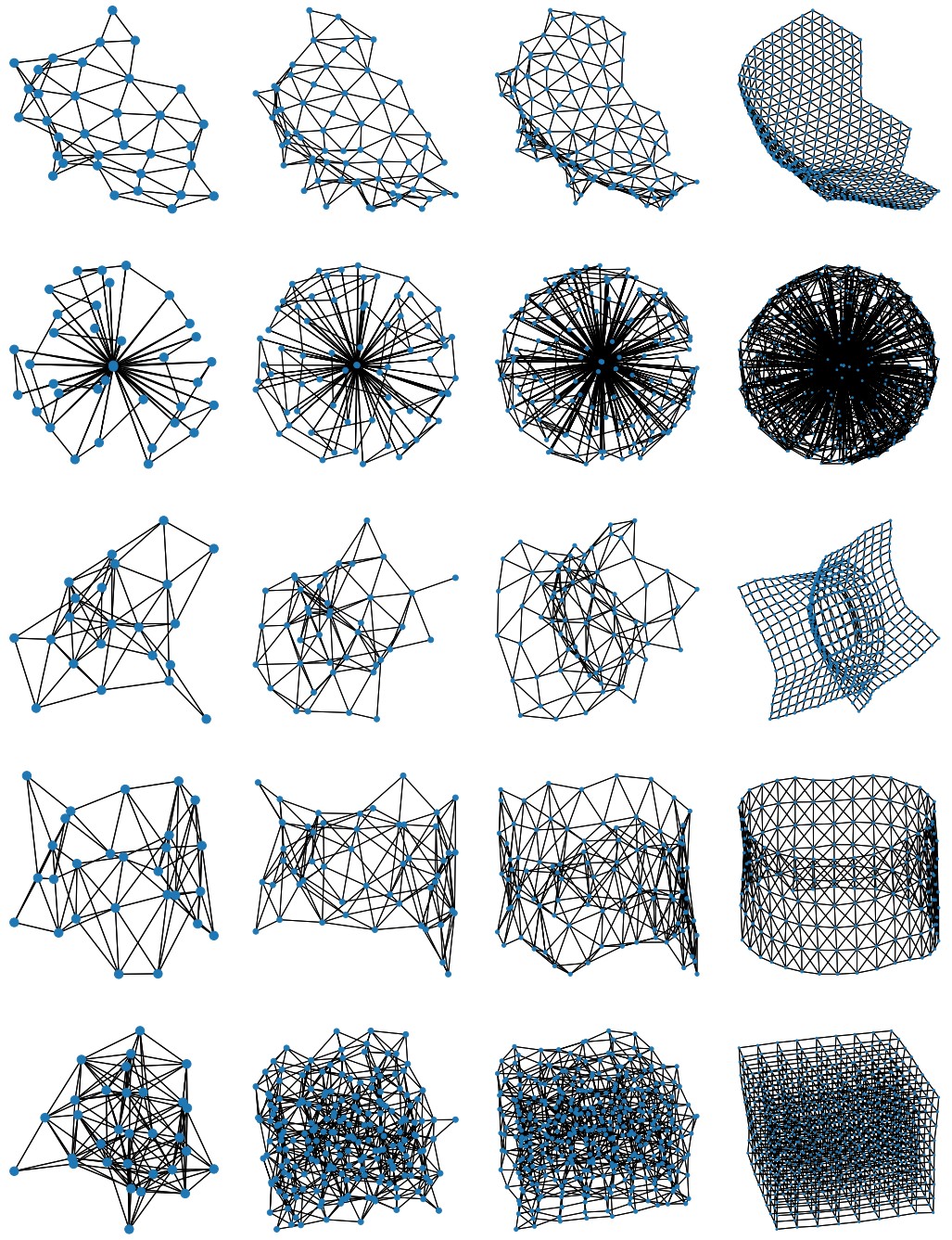

Figure 10: More examples of the evolution of CoRe-GD drawings. The model was trained on the suitesparse dataset and generates 3-dimensional drawings.

