# OpenReview forum: "CoRe-GD: A Hierarchical Framework for Scalable Graph Visualization with GNNs"
_ICLR.cc/2024/Conference — ICLR 2024 poster_

### Official Review · Reviewer_Te7K · 2023-10-27

**Soundness:** 3 good
**Presentation:** 2 fair
**Contribution:** 3 good
**Rating:** 6
**Confidence:** 3

**Summary:**

The manuscript presents a subquadratic graph drawing algorithm aimed at large-scale graph visualization. The authors propose a hierarchical optimization process to visualize node embeddings from coarse to fine. Additionally, a positional rewiring module is utilized to improve the connection between nodes with potential relationships. The experiments show that the proposed method not only accelerates computations but also improves the visualization quality.

**Strengths:**

* The paper is well-written with clear objectives and results.
* The algorithm detail is clearly defined and described.
* Alternative methods are discussed and compared for node initialization and positional rewiring step.
* The experimental section evaluates the state-of-the-art methods qualitatively and quantitatively.

**Weaknesses:**

* The motivation of the hierarchical optimization and positional rewiring is not clearly introduced. I would recommend the authors discuss the computational complexity among the proposed CoReGD, DeepGD, and SmartGD. In addition, the authors should introduce why positional rewiring is important in hierarchical optimization.
* I found it a bit hard to follow the method section. A diagram of the whole algorithm should be carefully introduced in this section, which can make the readers clearer to the proposed algorithm before diving into the details. It is difficult to understand the pipeline of CoReGD from Figure 2, e.g., h_1^1, h_1^l, h_1^{c+1}.
* I find the effectiveness of the proposed hierarchical optimization in Table 2 and Figure 6. However, the proposed positional rewiring, scale-invariant stress, and replay buffer do not make a significant impact on the stress metric from A.3 Table 3.
* More graph drawing quality metrics can be reported, such as shape-based metrics and edge crossing.

**Questions:**

Please see my comments to paper weakness.

---

> ### Author Response · Authors · 2023-11-14
>
> Thanks for taking the time to review our paper. We are addressing the weaknesses as follows:
>
> > The motivation of the hierarchical optimization and positional rewiring is not clearly introduced. I would recommend the authors discuss the computational complexity among the proposed CoReGD, DeepGD, and SmartGD. In addition, the authors should introduce why positional rewiring is important in hierarchical optimization.
>
> Thanks for the suggestion. We added Table 3 and a small paragraph to introduce the baselines and their computational complexity. We also improved Section 3 and now point out more clearly why the positional rewiring is useful.
>
> > I found it a bit hard to follow the method section. A diagram of the whole algorithm should be carefully introduced in this section, which can make the readers clearer to the proposed algorithm before diving into the details. It is difficult to understand the pipeline of CoReGD from Figure 2, e.g., h_1^1, h_1^l, h_1^{c+1}.
>
> We admit that Figure 2 could have been better for introducing the algorithm. We updated the manuscript and combined Figure 2 and 3 to serve the purpose you mentioned. We also added textual descriptions to make the overview more clear.
> The embeddings h_1^1, etc., are part of the figure as they also appear in the pseudocode in Appendix A.2. This makes it possible to align both the figure and the pseudocode to better understand how embeddings are passed exactly between modules.
>
> > I find the effectiveness of the proposed hierarchical optimization in Table 2 and Figure 6. However, the proposed positional rewiring, scale-invariant stress, and replay buffer do not make a significant impact on the stress metric from A.3 Table 3.
>
> Only looking at A.3 Table 3 might not show a big numerical difference in the ablation, but one has to keep in mind that the reported values are for an optimization problem that gets increasingly harder the closer one gets to the optimum. Referring back to Table 1 (which shows the base setup for the ablation study), we fall behind several other algorithms without positional rewiring, scale-invariant stress, or replay buffer. The numerical differences might seem small at first glance (for example, 236.65 instead of 233.17 without the replay buffer), but sgd^2 (233.49) and DeepGD (235.22) would be better without it. It is unclear where the optimum on the Rome dataset lies, but these components of our method are getting closer and closer to it.
>
> > More graph drawing quality metrics can be reported, such as shape-based metrics and edge crossing.
>
> We want to clarify that our framework is optimizing stress, and the positional rewiring is partly based on that assumption. Therefore, this is arguably the metric that needs to be used in the comparisons. However, optimizing other metrics could be interesting for future work.
>
> In conclusion, we address the mentioned weaknesses in the new revision of the paper, especially with the changes in Section 3. We would be very grateful if you could have a look at these changes. If there are any other questions, we are of course happy to help and would otherwise be happy if you could consider raising your score.

---

### Official Review · Reviewer_Avgq · 2023-11-01

**Soundness:** 3 good
**Presentation:** 3 good
**Contribution:** 3 good
**Rating:** 6
**Confidence:** 2

**Summary:**

The paper proposes CoreGD, which is a graph drawing framework which uses GNN submodules.  CoreGD improves over alternative methods in stress metrics and demonstrates empirically better scalability properties.

**Strengths:**

S1. The point about scale-invariant stress is nuanced and an astute observation -- the authors' lemma 2 helps support the scaling choice.

S2. Performance in graph drawing benchmarks appears to be quite strong in terms of stress metrics compared to alternative approaches.

S3. Authors were able to demonstrate better scaling performance than the next-best performing alternative (DeepGD it seems) in Figure 6 -- CoreGD seems to scale considerably better and with better stability.

**Weaknesses:**

W1. For the many readers unfamiliar with graph drawing benchmarks, it would be really helpful to have preliminaries as part of the main paper, and also discussions of the benchmarks to better understand the relative things being compared (I'd guess most GNN-familiar authors aren't familiar with graph drawing).

W2. An efficiency-stress curve would be great to understand relative runtimes and stress performances of the different methods in this landscape, as the authors' individual experiments suggest that CoreGD is the best performing and fastest.

**Questions:**

- 3.1 suggests the efficiency and effectiveness for CoreGD is largely influenced by the input feature choice.  Some of the features that offer high discriminability might be difficult/expensive to compute for large graphs (e.g. Laplacian PEs or beacons).

- The random noise addition step in 3.2 is interesting: this would means the output drawing is stochastic, I guess.  How sensitive are output drawing configurations to the noise introduced here?

- Section 3's overview of Core-GD is quite hard to understand.  Some design choices aren't obvious to the reader: (i) why do we need two convolutions (Conv_E and Conv_R), (ii) what's the purpose of the rewiring module?  How exactly does the GRU fit in?  Structuring this section with equations which the reader can follow from inputs to position coordinates would have probably helped understand the sequence of operations

- Table 1 suggests that Core-GD seems to achieve similar performance to DeepGD -- I'm not familiar with the differences with these methods, but some discussion regarding the speed advantages of CoreGD compared to DeepGD would be helpful to contextualize the added value of using the algorithm proposed by the authors.

- Being able to coarsen aggressively before initializing input features seems like a great advantage for large graphs -- I'm curious what the largest graph the authors were able to visualize with CoreGD is given this benefit.  The graphs in Table 1 seem to be mostly small -- are other graph drawing approaches (non-neural or neural) better able to support larger graphs?  In general, a runtime-stress curve which shows the relative performance of multiple methods would be really helpful to see where CoreGD lies on the frontier.

---

Updated my score after reading the authors' responses.

---

> ### Author Response · Authors · 2023-11-14
> **Rebuttal 1/2**
>
> Thank you for the review and the concrete proposals for improvements. We integrate them as follows:
>
> > W1. For the many readers unfamiliar with graph drawing benchmarks, it would be really helpful to have preliminaries as part of the main paper, and also discussions of the benchmarks to better understand the relative things being compared (I'd guess most GNN-familiar authors aren't familiar with graph drawing).
>
> Thanks for the suggestion. This is a good point. In reality, there is no good standardized way of benchmarking algorithms for Graph Drawing, especially for neural methods. The closest we get is the Rome dataset, which was also used by previous work, but these graphs are rather small and only stem from one task/distribution. This is why we add more datasets, some used for GNN benchmarking, bigger real-world instances from the suitesparse matrix collection, and synthetically generated sparse graphs. This makes our evaluation more extensive than previous work. We added a section describing our contribution regarding benchmarking in Appendix A.7.4 that we invite you to read. Beyond that, the first paragraph of Section 4 gives an overview of the employed datasets and why they were used.
>
> > W2. An efficiency-stress curve would be great to understand relative runtimes and stress performances of the different methods in this landscape, as the authors' individual experiments suggest that CoreGD is the best performing and fastest.
>
> We added plots in Appendix A.7.3 to show this. These show that CoRe-GD is indeed Pareto-optimal regarding runtime and layout quality.
>
> > 3.1 suggests the efficiency and effectiveness for CoreGD is largely influenced by the input feature choice. Some of the features that offer high discriminability might be difficult/expensive to compute for large graphs (e.g. Laplacian PEs or beacons).
>
> This is a good observation. First of all, we show that the features we use can be computed in sub-quadratic time. Of course, practical runtimes can still vary, but we want to point out that the computation for initial features only needs to be done for the coarsest graph, which we can bound by a maximum size (and thus assume to be constant). To back this up empirically, we compared both CoRe-GD with the coarsening hierarchy and without any coarsening in Figure 5. We can observe that for a graph size of around 12000, an inflection point is reached where the variant with coarsening becomes faster than the one without, exactly because the feature computation is becoming more and more expensive. As mention in the paper, this is one major advantage of the coarsening.
>
> > The random noise addition step in 3.2 is interesting: this would means the output drawing is stochastic, I guess. How sensitive are output drawing configurations to the noise introduced here?
>
> Indeed, all presented algorithms used as baselines in the paper produce randomized results. For CoRe-GD, this is also the case because it uses randomized features for nodes. To judge the impact of different training runs and executions, we provide the standard variation for all results in the paper. We observe that they are rather small and comparable to the classical algorithms.
> To further underline this, we computed the mean and standard deviation using only one model for datasets (which we provide with the code) over 5 runs. The results are as follows:
>
> | Rome  | Zinc  | MNIST  | CIFAR10  | PATTERN  | CLUSTER  | suitesparse |
> |---|---|---|---|---|---|---|
> | 233.25 $\pm$ 0.16 | 5.10 $\pm$ 0.02  |  129.07 $\pm$ 0.02 | 262.66 $\pm$ 0.06  | 3065.79 $\pm$ 0.05  |  2827.00 $\pm$ 0.02 | 42396.23 $\pm$ 66.12 |
>
> We can observe that the standard deviation is similar to or lower than when using multiple trained models. So, on one hand, the training is relatively stable and produces similar results over different training runs, and on the other, the trained models themselves show little variation when used multiple times with different random seeds.
> We want to point out that the drawings can still look different in ways that do not affect stress, e.g., the graph could be rotated. This is usually not a problem in practice and something that applies to all other methods, too.

---

> > ### Author Response · Authors · 2023-11-14
> > **Rebuttal 2/2**
> >
> > > Section 3's overview of Core-GD is quite hard to understand. Some design choices aren't obvious to the reader: (i) why do we need two convolutions (Conv_E and Conv_R), (ii) what's the purpose of the rewiring module? How exactly does the GRU fit in? Structuring this section with equations which the reader can follow from inputs to position coordinates would have probably helped understand the sequence of operations
> >
> > Thanks for pointing this out. We improved the writing in Section 3 and redesigned the overview figure to make this more clear. We also want to address the two questions here: (i) Conv_E and Conv_R work on different graph topologies, the first on the original graph and the second on the rewired edges. As these two have distinct characteristics, we decided to use two different parametrizations. (ii) The rewiring is done to let nodes that are not close to each other in the graph topology but close in the drawing directly exchange information with each other. This is important because nodes in close proximity (in the drawing) can have a large impact on the stress loss without knowing about each other. As we mention in the paper, previous work showed that the GRU works well for recurrent GNNs like ours with many applications of the same recurrent layer (possibly hundreds). The ablation study testing different convolutions in A.4 empirically validates this. Regarding formulas, we tried to pack the main part of the paper as densely as possible. Figure 2 contains the names of all embeddings used in the pseudocode we provide in Appendix A.2. This makes it possible to line up the figure with the pseudocode and helps to understand the sequence of operations from start to finish. A formula for the GRU convolution is also presented.
> >
> > > Table 1 suggests that Core-GD seems to achieve similar performance to DeepGD -- I'm not familiar with the differences with these methods, but some discussion regarding the speed advantages of CoreGD compared to DeepGD would be helpful to contextualize the added value of using the algorithm proposed by the authors.
> >
> > One main goal of CoRe-GD is to remain scalable by running in sub-quadratic time. This is the big distinguishing factor between CoRe-GD and DeepGD. To make this more clear, we added more discussion on runtimes of existing algorithms and other properties to Table 3 in the Appendix. As we show in our scaling study, the faster runtime of CoRe-GD allows it to run on graphs that are much bigger than those that DeepGD can handle.
> >
> > > Being able to coarsen aggressively before initializing input features seems like a great advantage for large graphs -- I'm curious what the largest graph the authors were able to visualize with CoreGD is given this benefit. The graphs in Table 1 seem to be mostly small -- are other graph drawing approaches (non-neural or neural) better able to support larger graphs? In general, a runtime-stress curve that shows the relative performance of multiple methods would be really helpful to see where CoreGD lies on the frontier.
> >
> > We added scatter plots for the runtime-stress comparison on different graph sizes in Figure 8. These show CoRe-GD as Pareto-optimal, meaning that it offers a worthwhile tradeoff. This holds especially when compared with neural methods, where it is both faster and provides higher-quality drawings. Regarding the graph sizes: It is true that Table 1 only depicts small graphs. Figure 5 shows the result of our scalability study with graphs up to size 25000 and the corresponding runtimes of all methods. (sgd)^2 and neato fall short when running on larger graphs (as well as DeepGD). PivotMDS is by far the fastest and thus very suitable for big instances but trades this off with worse quality.
> >
> > Overall, we are thankful for the questions and clarify them further in the revised version of the paper that was uploaded together with this comment. We thank the reviewer for helping us to strengthen the submission by providing these questions. If you feel that there are any other questions that are left unanswered or things we should clarify further, then we are very happy to help. We especially invite you to read the reworked Section 3 and would appreciate it if you could reconsider your score.

---

> ### Comment · Reviewer_Avgq · 2023-11-22
> **Thank you**
>
> Dear authors,
>
> Thanks for your detailed responses.  In particular, I'm pleased to see the stress-runtime plots demonstrating the competitiveness of the method, and also the (significant) rework of Section 3 which helps clarity.  I raised my score to a 6, which reflects my belief that this paper is above the bar.  I did not increase further because of my assessment about the scope of the work and potential for interest at ICLR.
>
> Best wishes,
>
> -Reviewer Avgq

---

### Official Review · Reviewer_GnFx · 2023-11-01

**Soundness:** 4 excellent
**Presentation:** 3 good
**Contribution:** 2 fair
**Rating:** 6
**Confidence:** 3

**Summary:**

CoRe-GD (CGD) is a effective novel graph drawing algorithm designed with scalability and computational efficiency in mind. The main block of CGD does recurrent layout optimization using two graph neural networks and a graph rewiring step. To achieve better scalability, CGD uses an iterative graph coarsening procedure. The procedure first learns a layout for the coarsest graph view and iteratively uncoarsens the graph and recomputes the optimal layout based on the prior, coarser graph layout.

Experiments show CGD is effective on an array of datasets with both real and synthetic graphs. Results also show that CGD is effective even on graph sizes larger than those seen during training.

CGD has scalability evidenced both in theoretical computational complexity as well as in experiments.

**Strengths:**

Originality - This work presents an original algorithm for learning graph drawing. The model also uses a recurrent GNN, enabled by memory replay. This itself is relatively original, though there are some similar related works.
Quality - This work is high quality. The model is clear and based on a solid foundation. The results and experiments show significant improvement.
Clarity - The work is clear, understandable, and well defined. Great clarity.
Significance - Outside of the Graph Drawing community, it is hard to see this having a broader impact. While the authors claim in the conclusion that because it is embedding based, CGD could be expanded to other applications, there is no direct evidence of that presented. Perhaps it could be used as positional embeddings in graph transformers.

**Weaknesses:**

I am not heavily knowledgeable about the Graph Drawing community and am not fully aware of the larger impacts (if any) outside of visualizations. It seems to me like this might be a relatively niche community but would defer to other reviewers on that front.

Otherwise, this is a pretty strong, high quality paper.

Figure 6 seems to show that CGD does worse than neato and (sgd)2 for the larger graphs. This is a weakness given that CGD is designed with larger scale graphs in mind.

neato and (sgd)2 do better on larger graphs and PivotMDS does almost as well on larger graphs but is much, much faster. This means CGD is only occupying a kind of middle ground on large graphs.

In addition, while I am not highly familiar with neato and (sgd)2, the fact that they are training free makes me believe they are less complex than CGD from an engineering perspective.

**Questions:**

“For a cycle graph, this is clearly not desirable and will end in a drawing that is far from optimal.” This is a good example. Would be good to have a visualization (maybe can go in appendix)

The figure 9 visualizations are a very nice way to see the effect of the graph coarsening.

Though I feel I understand it, a figure showing how the latent embeddings are transferred between supernodes and nodes in the uncoarsening step could be useful.

Why is DNN^2 excluded from figure 6?

Why does Figure 6.A cutoff at 5000 while 6.B cuts out at 25000?

While I believe I have a strong understanding of the algorithm presented, I do not have a good sense of this field and the most important features, datasets, and related work.

---

> ### Author Response · Authors · 2023-11-14
>
> We thank the reviewer for their honest review. We address the weaknesses as follows:
>
> > I am not heavily knowledgeable about the Graph Drawing community and am not fully aware of the larger impacts (if any) outside of visualizations. It seems to me like this might be a relatively niche community but would defer to other reviewers on that front.
>
> We agree that the main relevance lies in graph visualization. However, we believe that the methods we employed (adaptive positional rewiring, replay buffer with latent embeddings for training deep GNNs, etc.) are interesting for the wider graph learning community and can be of relevance for tasks beyond graph visualization.
>
> > Otherwise, this is a pretty strong, high quality paper.
>
> Thanks
>
> > Figure 6 seems to show that CGD does worse than neato and (sgd)2 for the larger graphs. This is a weakness given that CGD is designed with larger scale graphs in mind.
> neato and (sgd)2 do better on larger graphs and PivotMDS does almost as well on larger graphs but is much, much faster. This means CGD is only occupying a kind of middle ground on large graphs.
>
> There is a tradeoff between running time and quality for all algorithms. CoRe-GD beats all neural methods regarding speed and quality and overall offers a unique tradeoff between quality and computation speed when compared to all algorithms. To make this more clear, we added scatterplots depicting the runtime and quality of the algorithms for graphs of the same size (graphs of different sizes cannot be easily compared due to different optimal stress values) in Appendix A 7.3. We can observe that CoRe-GD appears Pareto-optimal.
>
> > In addition, while I am not highly familiar with neato and (sgd)2, the fact that they are training free makes me believe they are less complex than CGD from an engineering perspective.
>
> We are not 100% sure what you mean by "less complex" here, the strength of CoRe-GD is that it can be trained for different graph distributions without the necessity to adapt the employed heuristics of the algorithm (as they are learned and do not need human domain knowledge). One can easily retrain CoRe-GD with new graph distributions by using our existing code. This is not possible with the classical algorithms, where deliberate tuning of the heuristics is required to better fit a graph distribution. To demonstrate this adaptability, we use datasets from different domains in Table 1.
>
> > "For a cycle graph, this is clearly not desirable and will end in a drawing that is far from optimal." This is a good example. Would be good to have a visualization (maybe can go in appendix)
>
> Thanks for the suggestion. For a cycle graph, this would lead to all nodes being positioned on the same point, which is clearly not stress optimal (as mentioned in the paper). We can add a visualization with a cycle graph and a single node representing the resulting drawing next to it, but we don't feel that this adds much to the paper. Is there another visualization that you have in mind?
>
> > Though I feel I understand it, a figure showing how the latent embeddings are transferred between supernodes and nodes in the uncoarsening step could be useful.
>
> We generally improved the figure for the architecture by combining the old Figures 2 and 3. While doing so, we added more depictions of the graph while being processed (also one before and after the uncoarsening) and added more descriptions. We hope that this helps.
>
> > Why is DNN^2 excluded from figure 6?
>
> The training of DNN^2 was extremely unstable and did not converge for bigger graphs (leading to huge losses). This is also why standard deviations for DNN^2 on the small datasets are by far the biggest. As the performance of DNN^2 is already severely lacking behind on small graphs, we do not see this as a significant problem. A remark on this is also contained in the paper.
>
> > Why does Figure 6.A cutoff at 5000 while 6.B cuts out at 25000?
>
> The difference between plot 6.A and 6.B is that for 6.A, we have to compute losses, which is not necessary for 6.B (as it only measures inference runtimes). To compute the full stress loss, we have to compute all-pairs-shortest paths and compare them to the distance in the drawing. This becomes prohibitively costly in both runtime and memory requirements, which is why we had to cap the graph size for Figure 6.A. We also mention this in the paper.

---

> > ### Author Response · Authors · 2023-11-14
> >
> > > While I believe I have a strong understanding of the algorithm presented, I do not have a good sense of this field and the most important features, datasets, and related work.
> >
> > Looking at existing work, the evaluation for graph drawing is not as standardized as one would hope, and neural methods usually only use the Rome dataset for their comparisons. We add five more datasets of similar sizes from different domains and add both graphs from the suitesparse matrix collection and synthetic sparse graphs. We added more background information on benchmarking for Graph Drawing in Appendix A.7.4, which we invite you to read.
> >
> > Again, we thank you for approaching our paper with an open mind and posing relevant and interesting questions. If you feel that we addressed your questions adequately, then we would be happy if you could consider raising your score.

---

### Official Review · Reviewer_hLpG · 2023-11-01

**Soundness:** 3 good
**Presentation:** 3 good
**Contribution:** 3 good
**Rating:** 5
**Confidence:** 3

**Summary:**

Stress is a widely used metric for graph visualization, which aims to find geometric embeddings of graphs that optimize certain criteria. As the stress optimization presents computional challenges due to its inherent complexity and is usually solved using heuristics in practice. The authors introduce a scalable Graph Neural Network based Graph Drawing framework with sub-quadratic runtime that can learn to optimize stress. Inspires by classical stress optimization techniques and force-directed layout algorithms, they create a coarsening hierarchy for the input graph. Beginning at the coarsest level, they iteratively refine and un-coarsen the layout, until generating an embedding for the original graph. The authors perform empirical evaluation demonstrating that their framework achieves SOTA while remaining scalable.

**Strengths:**

1.	Extensive examples, clear explanations,  step-by-step formulations, and open-sourced code make this research solid, convincing and reproducible. The presentation is obviously above the average.
2.	Extensive experiments demonstrate the effectiveness and the scalability of the proposed framework.
3.	This research provide insights about cobining the GNN with the graph combinational optimization (i.e. graph visualization). I believe this paper would benifit to the graph mining community.

**Weaknesses:**

1. Some training details are put in the appendix. I found that after reading the main body of the paper, I couldn’t understand how the framework is applied for training and serving. I think the writing order of the paper needed to be improved. And the figure 2 is relatively hard to understand. Please give a more clear illustration about the graph after being processed by each module in your framework.

2. The work is related with graph condensation or graph summarization, and some baselines about graph condensation or summarizarion should be included.

**Questions:**

1. Please give a more clear illustration about the graph after being processed by each module in your framework.
2. Is this research related with graph condensation or graph summarization? I mean, maybe some baselines about graph condensation or summarizarion should be included if so.

---

> ### Author Response · Authors · 2023-11-14
>
> We want to thank the reviewer for their review and the questions they brought up. We address the two weaknesses/questions as follows:
>
> > Some training details are put in the appendix. I found that after reading the main body of the paper, I couldn't understand how the framework is applied for training and serving. I think the writing order of the paper needed to be improved. And the figure 2 is relatively hard to understand. Please give a more clear illustration about the graph after being processed by each module in your framework.
>
> Thanks for the suggestion. We combined Figures 2 and 3 and gave them an overhaul with newly added textual descriptions and more depictions of the graph between the different processing steps to make them easier to understand. The writing in Section 3 has also been improved to make the execution clear. The main body now gives a clear overview of the algorithm, while detailed pseudocode for both training and inference is given in Appendix A.2.
>
> > Is this research related with graph condensation or graph summarization? I mean, maybe some baselines about graph condensation or summarizarion should be included if so.
>
> Graph condensation and summarization (in the context of graph learning, we assume this is what you refer to, please correct us if we are wrong) are concerned with reducing the size of a graph while maintaining important information, usually based on label information and the graph structure. As a byproduct of the graph being reduced, this might lead to more pleasing layouts once visualized. On the other hand, the graph drawing task we are considering finds a stress-optimized spatial embedding for the original graph. These tasks are inherently different, and we do not see how a meaningful comparison with baselines for graph condensation and summarization to our method can be done.
> We could also not find more information on a further relation between graph drawing and condensation/summarization in the literature. If there is a comparison that you have in mind which would make sense for our task, then please let us know. Otherwise, if this was your main concern, then we would be happy if you would consider increasing your score.

---

### Author Response · Authors · 2023-11-21
**Global response**

Dear reviewers,

We want to express our gratitude for your time and dedication in reviewing our work. As we still await your feedback on our rebuttal, we are keen on engaging with you to discuss any open questions.
We are pleased to highlight that the majority of the feedback received thus far has been positive. Noteworthy is the constructive feedback that prompted clarifications, all of which have been incorporated into the revised paper (with changes in red). Allow us to outline the key enhancements again:
* **Figure 2:** We have improved Figure 2 to provide a clearer explanation of the framework's steps. Additional example graphs have been included to better illustrate the impact of the modules. In addition to the overview, the figure contains descriptors for the embeddings that are consistent with their use in Algorithm 1, making it easier to understand all processing steps in detail.
* **Section 3 Writing:** We have improved the clarity of Section 3, and now provide a better motivation for the use of positional encodings. We also refer to the newly added Figure 6 in the Appendix for additional context.
* **Appendix A.7.3 - Runtime/Stress Plots:** In response to your feedback, we have added runtime/stress plots to Appendix A.7.3, offering insights into CoRe-GD's tradeoff between runtime and solution quality. Our observations indicate Pareto-optimal results.
* **More Background on Benchmarking:** We have added more details on benchmarking in Appendix A.7.4, particularly for readers unfamiliar with Graph Drawing. Not only do we include classical datasets (Rome and suitesparse) for comparability, but also expand the evaluation with five additional datasets from the Graph Learning community and synthetically generated sparse graphs for our scalability study.

We believe these changes enhance the clarity of the paper and provide additional background information and motivation. We are eager to hear if there are any further issues or concerns that need attention. If this is not the case, we kindly ask you to consider changing your score accordingly.

Best regards,

The authors

---

### Meta-Review · Area_Chair_ZWrV · 2024-01-07

**Metareview:**

The author response managed to convince most reviewers that this paper is above the bar. It is an unusual application of GNNs which could actually make the paper more interesting for the ICLR audience.

**Justification For Why Not Higher Score:**

low confidence reviews

**Justification For Why Not Lower Score:**

most concerns addressed

---

### Decision · Program_Chairs · 2024-01-16

Accept (poster)